



# The Regional Coupled Suite (RCS-IND1): application of a flexible regional coupled modelling framework to the Indian region at km-scale

Juan M. Castillo[1], Huw W. Lewis[1], Akhilesh Mishra[2], Ashis Mitra[2], Jeff Polton[3], Ashley Brereton[3], Andrew Saulter[1], Alex Arnold[1], Segolene Berthou[1], Douglas Clark[4], Julia Crook[5], Ananda Das[6], John Edwards[1], Xiangbo Feng[7], Ankur Gupta[2], Sudheer Joseph[8], Nicholas Klingaman[7], Imranali Momin[2], Christine Pequignet[1], Claudio Sanchez[1], Jennifer Saxby[5], Maria Valdivieso da Costa[7]

[1]Met Office, Exeter, EX1 3PB, UK
[2]National Centre for Medium Range Weather Forecasting (NCMRWF), India
[3]National Oceanography Centre, Liverpool, UK
[4]UK Centre of Ecology & Hydrology (UKCEH), UK
[5]University of Leeds, UK
[6]India Meteorological Department (IMD), India
[7]University of Reading, UK
[8]INCOIS, India

*Correspondence to*: Juan M. Castillo (juan.m.castillo@metoffice.gov.uk)

**Abstract.**

A new regional coupled modelling framework is introduced - the Regional Coupled Suite (RCS). This provides a flexible research capability with which to study the interactions between atmosphere, land, ocean and wave processes resolved at km-scale, and the effect of environmental feedbacks on the evolution and impacts of multi-hazard weather events. A configuration of the RCS focussed on the Indian region, termed RCS-IND1, is introduced. RCS-IND1 includes a regional configuration of the Unified Model (UM) atmosphere, directly coupled to the JULES land surface model, on a grid with horizontal spacing of 4.4 km, enabling convection to be explicitly simulated. These are coupled through OASIS3-MCT libraries to 2.2 km grid NEMO ocean and WAVEWATCH III wave model configurations. To examine a potential approach to reduce computation cost, and simplify ocean initialisation, the RCS includes an alternative approach to couple the atmosphere to a 1ower resolution Multi-Column K Profile Parameterization (KPP) for the ocean. Through development of a flexible modelling framework, a variety of fully and partially coupled experiments can be defined, along with traceable uncoupled simulations and options to use external input forcing in place of missing coupled components. This offers a wide scope to researchers designing sensitivity and case study assessments. Case study results are presented and assessed to demonstrate the application of RCS-IND1 to simulate two tropical cyclone cases which developed in the Bay of Bengal, namely Titli in October 2018 and Fani in April 2019. Results show realistic cyclone simulations, and that coupling can improve the cyclone track and produces more realistic intensification than uncoupled simulations for Titli but prevents sufficient intensification for Fani. Atmosphere-only UM regional simulations omit the influence of frictional heating on the





boundary layer to prevent cyclone over-intensification. However, it is shown that this term can improve coupled simulations,
enabling a more rigorous treatment of the near-surface energy budget to be represented. For these cases, a 1D mixed layer
scheme shows similar first-order SST cooling and feedback on the cyclones as a 3D ocean. Nevertheless, the 3D ocean
generally shows stronger localised cooling than the 1D ocean. Coupling with the waves have limited feedback on the
atmosphere for these cases. Priorities for future model development are discussed.

## 1 Introduction

There is a growing focus from researchers around the world on the potential of more integrated coupled approaches to
environmental prediction on regional scales. A key driver for this development is to provide more accurate forecasts and
warning of natural hazards and their impacts, focusing where multiple hazards occur concurrently and where representing
the effect of air-sea interactions impacts the evolution of high impact weather systems. The application of regional coupled
models is gaining attention to improve simulations focussed on both short-term operational natural hazard prediction (e.g.
Ruti et al., 2020) and longer timescale assessments of environmental change (e.g. Gutowski et al., 2020).

This paper describes a new km-scale atmosphere-land-ocean-wave coupled system designed to support research on the
sensitivity of environmental predictions in the Indian region to the representation of interactions and feedbacks between
model components. As reviewed by Hagos et al. (2020), the application of atmosphere-ocean coupled systems in tropical
regions is particularly relevant given that air-sea interactions drive and can impact tropical meteorological processes. The
modelling framework described in this paper is defined to run at km-scale across all components, to enable explicit
representation of key processes including atmospheric convection (e.g. Turner et al., 2019; Volonté et al., 2020) and ocean
eddies and internal tides within shelf seas (e.g. Jithin et al., 2019). This resolution also offers the potential to represent
catchment-scale hydrology and land-sea interactions at coastlines with better fidelity than typically possible with regional
and global-scale coupled model approaches running with grid resolutions of order 10 km or coarser (e.g. Eilander et al.,
2020).

Recent studies have highlighted the sensitivity of environmental processes in the Indian region to interactions between
different components of the environment. For example, Roman-Stork et al. (2020) examined reanalysis products to
demonstrate that reduced transport of fresher water from the Bay of Bengal over the past 15 years, fed by river discharge,
has increased the depth of a barrier layer in the south-eastern Arabian Sea, in turn contributing to a reduction in the number
of intense monsoons. Salinity – precipitation feedback mechanisms were also explored by Krishnamohan et al. (2019), who
focused on more localised processes in the Bay of Bengal. Karmakar and Misra (2020) found propagation of the summer
monsoon rainfall to be faster over the Arabian Sea than Bay of Bengal due to a relative enhancement of convection over the
Arabian Sea associated with moisture convergence.





On shorter timescales, the importance of air-sea interaction in modulating the evolution of tropical weather systems is also well known. This is most clearly illustrated for tropical cyclones (TC) that are prevalent within the Bay of Bengal, and there is a notably high number of studies published on this topic in recent years. For example, TC Vardah (December 2016) was shown to be sensitive to atmosphere-wave coupling and the inclusion dynamic sea spray flux in the COAWST system of Warner et al. (2010) (Prakash et al. 2019) and to atmosphere-ocean mixed layer coupling, with sensitivity depending on initial mixed layer depth (Yesubabu et al. 2019). The sensitivity of TC simulations using a regional coupled model were found to be highly sensitive to the surface drag parameterisation by Greeshma et al. (2019). Mohanty et al. (2019) quantified improvements in TC position and timing errors of 20% and 33% respectively for HWRF (Biswas et al., 2018) atmosphere model simulations of several TCs in the Bay of Bengal when applying 6-hourly updating SST boundary conditions compared to a control simulation in which the SST conditions are persisted through the simulations. This latter approach is typical of regional modelling configurations used in most operational NWP centres (e.g. Routray et al., 2017; Mahmood et al., 2021). The relevance of spatial resolution of SST boundary conditions for atmosphere-only simulation of TC Phailin was examined by Rai et al. (2018), who found improved performance by order 30-40% when using a higher resolution (0.083° × 0.083°) SST analysis.

Beyond the well documented impacts of TC multi-hazards on lives and livelihoods (e.g. Pandey et al., 2021), the impact of TCs on physical and biogeochemical ocean processes in the Bay of Bengal are also receiving increasing attention. For example, Maneesha et al. (2019) and Girishkumar et al. (2019) discussed the observed impact of TC Hudhud on upper ocean dynamics and chlorophyll, finding this was maintained for two weeks after the passage of the storm. The impact of TC Phailin on the upper ocean was assessed by Jyothi et al. (2019) and Qiu et al. (2019) in different ocean-only simulations, building on the earlier COAWST coupled model assessment of this case by Prakash and Pant (2017). The signature cooling of the ocean mixed layer by as much as 7 °C in response to the passage of the TC was noted, in addition to strong TC-induced upwelling and substantial increases of up to 5 psu in surface salinity over these regions. Maneesha et al. (2021) highlighted the considerable impacts that TCs can have on marine biogeochemistry in the region but noted relatively limited impacts of TC Titli due to persistent stratification in western regions suppressing upwelling.

This paper aims to document the first implementation of a regional coupled modelling system focussed on the Indian region that uses the Unified Model atmosphere, JULES land surface, NEMO or Multi-Column K Profile Parameterization (KPP) ocean, and WAVEWATCH III wave model codes. These are run on grids with horizontal spacing or order a few km (4.4 km atmosphere, 2.2 km ocean), which enables explicit representation of atmospheric convection, ocean eddies and internal tides and their interactions. This represents a marked increase in spatial resolution relative to most coupled modelling studies highlighted above for the region that tend to be based on atmosphere (typically WRF; Skamarock et al., 2008) and ocean (typically ROMS; Shchepetkin et al, 2005) simulations running at order 10 km resolution or coarser, for which atmospheric





convection is parameterised. Furthermore, there has been relatively little assessment of the role of wave processes in modifying the air-sea interactions under extreme conditions such as TCs in previous modelling studies. The option of using a lower horizontal resolution KPP ocean mixed layer parameterisation component is also introduced here to examine the performance of a computationally cheaper coupled configuration relative to the 2.2 km resolution 3D general ocean circulation model with a full dynamical ocean representation. This paper is organised as follows. Section 2 describes the

RCS-IND1 definition of the RCS modelling framework and its component configurations. Section 3 provides an initial assessment of system performance and the impact of coupling for case study simulations of cyclones Titli and Fani which developed in the Bay of Bengal during October 2018 (post-monsoon) and April 2019 (pre-monsoon) respectively. Future development priorities are finally outlined in Section 4.

## 2 The RCS-IND1 implementation of the Regional Coupled Suite prediction framework

The first version of the India-focussed implementation of the RCS, termed RCS-IND1 for brevity, builds on the development of a regional environmental prediction system using the same model components and grid resolutions focussed on the north-west European shelf region (UKC; Lewis et al., 2019; Lewis et al., 2018). RCS-IND1 incorporates atmosphere, land surface, ocean and wave model components, with coupling between components where required achieved with the OASIS3-MCT (Ocean Atmosphere Sea Ice Soil) coupling libraries (version 2.0; Valcke et al., 2015). Configurations of the following model

codes are included:

- Unified Model (UM) atmosphere (version 11.1; e.g. Brown et al., 2012),
- Joint UK Land Environment Simulation (JULES) land surface model (version 5.2; Best et al., 2011; Clark et al., 2011),
- Nucleus for European Models of the Ocean (NEMO) 3D ocean model (version 4.0.1; NEMO team, 2019),

- Multi-Column K Profile Parameterization (KPP) 1D ocean model (version 1.0; Hirons et al., 2015)
- WAVEWATCH III surface wave model (version 4.18; WW3DG, 2016).

As described below, some code modifications are applied to the referenced versions for use in the RCS-IND1 configuration, mainly related to optimization, introduction of coupling capability, or to enable additional diagnostic output. The UM and JULES models run together as one single executable, and therefore no OASIS-MCT coupler is needed to exchange

information between them.

### 2.1 Model domain and grids

The model domain illustrated in Fig. 1 covers the region 3.5ºN - 40ºN and 65ºE - 101ºE. This is selected to be comparable to the extent and grid resolution of the NCUM-R operational atmosphere configuration (Jayakumar et al., 2017; Mamgain et al., 2018; Jayakumar et al., 2019) and operational coastal ocean and wave modelling capabilities (e.g. Francis et al., 2020;

Remya et al., 2020). The atmosphere and land components have a fixed resolution latitude-longitude grid with horizontal





grid spacing of 4.4 km, which translates to 900 grid cells in the west-east zonal direction and 904 in the north-south meridional direction. The benefit of building on the NCUM-R domain is that potential atmosphere model issues, for example those related to steep Himalayan orography, have previously been considered and addressed. Relative to the NCUM-R atmosphere domain, the RCS-IND1 coupled system domain is marginally extended to the East to cover the whole ocean

extent of the Malacca strait, and to the South to better capture more of the equatorial ocean currents towards the southern boundary.

The ocean and wave components extend across all sea areas of the coupled model domain (Figure 1). The NEMO ocean component is defined on a fixed resolution grid with a horizontal resolution of 2.2 km (1760 grid cells in the zonal and 1100

in the meridional directions). The wave model uses a Spherical Multiple-Cell (SMC) grid (Li, 2011) with 425841 unstaggered wave grid cells with grid spacing of 2.2 km in coastal areas and expanding to 4.4 km in open seas. Note that sea grid points in the Gulf of Thailand to the far east of the domain are masked in the ocean and removed in the wave model, so that ocean and wave calculations do not take place in this region. The KPP mixed-layer parameterisation component is applied on the same domain, but with a coarser horizontal grid with spacing of 15 km to match the initial ocean analysis

resolution (262 grid cells in the zonal and 392 in the meridional directions), and apply the 1D scheme at sufficiently coarse resolution that horizontal advection can be neglected.

## 2.2 Coupling framework

The RCS is built as a rose suite (http://metomi.github.io/rose/doc/html/index.html, last access: 27 October 2021) that defines the component configurations and methods for running simulations. Models are run with daily cycling, whereby every cycle

after the first day uses the final state of the previous cycle to re-initialize. Details of component initialisation are provided further below. The flexibility of the RCS results from it being possible to run one of multiple different coupled and uncoupled configuration options (Figure 2), based on the setting of a *RUN_MODE* environment variable specified from within the same rose suite (see Table 1 and Table 2). This flexibility prevents the need to develop and maintain different suites for different run options (e.g. as in Lewis et al., 2018; Lewis et al., 2019). The main modes of running simulations are:

- Fully coupled (***RUN_MODE = atm-ocn-wav***): two-way feedbacks represented between all model components in the system (Table 1).
   - Partially coupled (***RUN_MODE = atm-ocn, atm-kpp, atm-wav, or ocn-wav***): two-way feedbacks represented between only two selected components (Table 1).
   - Uncoupled (***RUN_MODE = atm, ocn, or wav***): no coupled feedbacks with external components are represented,

but model components can be configured with different forcing options (Table 2).

Surface boundary conditions are provided via file forcing when not available via coupling, such as when running in partially coupled or uncoupled mode. Different choices for the source of file forcing data can be specified in the RCS as environment



variables prior to a model run (see Tables 3 – 5), which are pre-processed as part of the suite workflow during run time given
the location of the source data.

The KPP ocean component is only currently supported in the suite to run in atmosphere-KPP coupled mode, and therefore no
additional forcing is necessary. Table 1 and Table 2 summarize all currently available research configurations of the RCS
modelling framework, and illustrated in Figure 2, along with the associated naming convention introduced to support a
variety of potential experimental designs. Not all possible configurations will be further discussed in this paper for
simplicity.

Namelists defining all the available RCS-IND1 configurations are provided under the rose framework as suite *u-bf945*,
accessible to registered researchers under a repository at https://code.metoffice.gov.uk/trac/roses-u/browser/b/f/9/4/5/trunk
(last access: 27 October 2021). A more detailed description of the namelists used for each configuration is included in the
Supplement to this paper.

The exchange of model variables between each coupled component and their required order of coupling (Table 3) have
previously been described in detail for the UKC2 and UKC3 regional coupled systems (Lewis et al., 2018; Lewis et al.,
2019), although there are some minor differences in the fields that are now exchanged. When coupling the ocean to the wave
model (with or without coupling to the atmosphere model), components of the water-side stress vector transmitted into the
ocean are exchanged from the wave to the ocean model, rather than the previous approach where the fraction of the
atmospheric momentum transferred to the ocean was exchanged. The surface momentum budget can be expressed as:

$$\tau_{oc} = \tau_{atm} - \tau_{wav} + \tau_{wav:ocn} \quad , \qquad (1)$$

where $\boldsymbol{\tau}_{oc}$ is the atmospheric stress transmitted into the ocean, $\boldsymbol{\tau}_{atm}$ the stress applied by the atmosphere on the ocean surface,
$\boldsymbol{\tau}_{wav}$ the momentum flux absorbed by the wave field, and $\boldsymbol{\tau}_{wav:ocn}$ the momentum stored by waves that is transferred to the
ocean through wave breaking. The advantage of exchanging components rather than a fractional momentum transfer is that
the atmospheric stress transmitted into the ocean does not diverge when the stress applied by the atmosphere on the ocean
surface is very small, and it is applied in the correct direction in all cases. When the ocean is coupled only to the atmosphere
(no wave coupling), the water-side stress transmitted into the ocean is equal to the atmospheric momentum. The wind speed
defined at 10 metres above the ocean surface is no longer exchanged, and this parameter is only used in the ocean model
when forced using the bulk formulation. Finally, although it is possible to pass the local water depth from the ocean to the
wave model, this is not done in RCS-IND1, as it was found that extensive changes to the wave code running for SMC wave
grids would be required to enable this exchange. This issue will be revisited in future in the context of updating the
WAVEWATCH III code.





All coupling fields are computed as hourly mean values and exchanged every hour of the simulation starting from the
initialization of the models (time step zero). A series of experiments using the UKC3 mid-latitude domain determined that
increasing the coupling frequency from an hour to every 10 min did not substantially impact results in that domain, but this
will need to be revisited for the RCS-IND1 tropical domain to better represent timescales for changing conditions in squalls
and tropical cyclones.

### 2.3 Atmosphere and land surface components

The atmosphere and land surface components in RCS-IND1 use the RAL1-T science configuration, for which science
parameters and performance are described in detail by Bush et al. (2020). RAL1-T uses an 80-level terrain-following vertical
coordinate set with Arakawa C-grid staggering (Arakawa and Lamb, 1977), up to 40 km altitude with 59 levels in the
troposphere below 18 km and 21 levels further above. The land surface is defined with 4 soil levels to a depth of 3 m, and 9
land surface tiles to represent land surface heterogeneity within each 4.4 km wide grid cell. The integration time step is set to
120 seconds.

The initial state is taken from reconfiguration (interpolation) of a global-scale UM analysis for a given initialisation time. For
the experiments described in this paper, these are provided by the operational analysis used for Met Office global numerical
weather prediction (Walters et al., 2019; Sharma et al., 2021), which was running with a horizontal grid resolution of order
17 km at tropical latitudes at the time of Titli and Fani cases. Horizontal boundary conditions are provided by re-running
simulations of that global UM configuration. Given the extended length of case study simulations considered here, those
global UM simulations were re-initialised from a new analysis each day through a case study, such that lateral boundary
conditions were no more than 24 h beyond a new analysis time.

Optimization and coupling modifications were added to the UM version 11.1 reference code for RCS-IND1, in order to:
- Activate wave coupling capabilities independently of ocean coupling.
- Add OMP barriers (OpenMP) to avoid threads accessing the same memory without proper synchronization (data race).
- Adjust the bounds of some loops and coordinate write sentences between OMP threads.

The following code changes were also introduced in JULES version 5.2:
- Introduction of a variable Charnock parameter at each grid point.
- Improve the convergence stability in the calculation of the Obukhov length.





When run without ocean coupling, different approaches are supported in the RCS to define how the sea surface temperature (SST) is applied as a lower boundary condition to the atmosphere component, controlled by *SST_INIT* and *SST_REINIT* environment variables (Table 4):

- Initial condition SST read either from a global UM analysis (*SST_INIT = none*), which uses the OSTIA analysis (Donlon et al., 2011) available at the time of its creation and mapped onto the global UM grid or from reading
OSTIA data directly on its native 0.05° grid (*SST_INIT = ostia*). This initial condition SST is either kept constant for the duration of a simulation (the default) or can be updated daily through the run (set if *SST_REINIT = true*).

- Initial condition SST interpolated from a km-scale resolution ocean model simulation data, e.g., the output of an ocean-only simulation of the RCS-IND1 (*SST_INIT = high_sst*; note not applied in this paper), and then either kept fixed or updated hourly through the run if *SST_REINIT = true*.

When running in uncoupled mode, surface ocean currents are assumed to be zero, and the Charnock parameter has a constant value at all ocean points of 0.011.

**2.4 NEMO ocean component**

The ocean component in RCS-IND1 is defined with the same science configuration as used in the AMM15 ocean model developed initially for the North-west European shelf region (Graham et al., 2018; Tonani et al., 2019) and applied in the
UKC regional coupled system (Lewis et al., 2019). AMM15 runs at a similar 1.5 km eddy-permitting horizontal resolution to the 2.2 km grid used for the Indian region. Some changes were applied relative to that configuration due to both updating the NEMO version from 3.6 to 4.01 and to attempt to account for specific details of the India domain:

- Increase the integration time step from 60 to 90 seconds. This is possible because the lower grid resolution of the RCS-IND1 model relaxes the stability conditions relative to AMM15.
- In uncoupled or ocean-wave simulations using the bulk formulation for atmospheric forcing, the Large and Yeager (2009) algorithm is substituted by the COARE 3.5 algorithm (Edson et al. 2013), as it is closer to the formulation that is used in operational implementation of AMM15 (Tonani et al., 2019).

- The formulation of the momentum advection changes from the vector form second centred scheme to the flux form third order UBS scheme (Shchepetkin and McWilliams, 2005), as the former scheme will be removed in later
versions of NEMO.

- For the RCS-IND1 configuration only one set of ocean boundary conditions is needed.

- Tidal data is read only on boundary segments, instead of assuming that each tidal data file contains all complex harmonic amplitudes.

The ocean model component has 75 vertical levels with a vertical grid using a hybrid terrain following z-s coordinate system (NEMO Team, 2019), and a non-linear free surface condition. The ocean bathymetry at 2.2 km resolution is based on the 2-



Minute Gridded Global Relief Data (ETOPO2), modified to improve shallow regions (Sindhu et al., 2007). In the initial configuration described here, no river forcing is applied, which is recognised will compromise the quality of simulated ocean salinity structures. In future, it is envisaged that river flows simulated from the land component will feed into the ocean (e.g. Lewis and Dadson, 2021).

Several options for forcing the NEMO ocean model are supported when running without atmosphere coupling (Table 5):

- bulk formulation with ERA5 (Hersbach et at., 2019) input data (*UM_FORCING = file-core*),
- flux formulation using global atmosphere model data (*UM_FORCING = flx-global*),
- flux formulation using km-scale resolution regional atmosphere data, such as the data produced by an atmosphere-only simulation of the RCS-IND1 configuration (*UM_FORCING = flx-high*).

Ocean-forced runs using the flux formulation are more easily comparable to ocean coupled runs, as the surface boundary condition forcing fields are equivalent to the coupling fields (Table 3). For more detail on the differences between the bulk and the flux forcing formulation, see NEMO team (2019).

For the case studies presented in this paper, the 3D ocean state is initialized by a multi-annual RCS-IND1 ocean-only simulation with ERA5 forcing starting from rest conditions on the $1^{st}$ of January 2016, where the initial temperature and salinity profiles were obtained from the Global_Analysis_Forecast_PHY_001_024 product of the Copernicus Marine Environment Monitoring Service (CMEMS), available after registration to the Copernicus services on http://marine.copernicus.eu/ (last accessed 27 October 2021). The initial temperature and salinity profiles were horizontally and vertically interpolated to the 2.2 km model grid beforehand, with coastal areas inundated and steep horizontal gradients smoothed via linear interpolation to maintain the initial stability of the run. Daily updated horizontal boundary data were obtained from the same global CMEMS product, where horizontal interpolation to the model grid is required prior to running simulations, but vertical interpolation is applied 'on the fly' during simulation by a modification to the NEMO code. The rim width of the boundary data is 9 grid cells, compared to 15 in the AMM15 configuration. Tidal forcing at open boundaries takes place as a sum of five tidal constituents (M2, S2, K2, O1, K1) obtained from the FES2014 tidal model, available after registration on https://datastore.cls.fr/catalogues/fes2014-tide-model/ (last accessed 27 October 2021). Further details and guidance on the workflow for the generation of the ocean component is provided by Polton et al. (2020).

Some modifications to the NEMO version 4.01 trunk source code have been made to correct issues or enable additional capabilities, as follows:

- Use the mean sea level pressure value obtained via coupling when available, instead of using a forcing file.
- Compute additional mixed layer depth diagnostics.
- Perform coupling exchanges before the initial time step of the simulation, so that the initial values of the coupling fields are available at the beginning of the run.





- Amend vertical interpolation of boundary data when the number of levels of the input boundary data is not the same as the number of levels of the model.
- Add a time stamp in the NEMO restart file name for convenience during post processing.

**2.5 Multi-Column K Profile Parameterization (KPP) ocean component**

The multi-column KPP version 1.0 code (Hirons et al., 2015) was used without modifications to couple to the atmospheric model. The same regional configuration as described by Klingaman and Woolnough (2014) for the Indo-Pacific region was used in this study. By using the vertical mixing scheme of Large et al., (1994), KPP provides a computationally efficient approach to simulate one-dimensional processes such as heat fluxes in the vertical, and re-distribution within the water column in the absence of horizontal advection processes. Initial conditions for temperature, salinity, and current velocity
components were obtained via vertical and horizontal interpolation of the operational ORCA025 global ocean model analysis run at the Met Office (Blockley et al., 2014). Over the India-focussed domain used in this configuration, the KPP component has a latitude-longitude horizontal grid with 0.094° latitude and 0.141° longitude spacing (order 15 km resolution) and 100 vertical levels.

**2.6 WAVEWATCH III surface wave component**

The surface wave model component uses WAVEWATCH III version 4.18, with some modifications applied for RCS-IND1 to improve the support for coupling all required fields. The component applied in RCS-IND1 is the same as for the UKC3 regional coupled environmental prediction system (Lewis et al., 2019) with minor improvements detailed below:

- Minor bug fixes for declaration of constant variables,
- Improved initialization of coupling fields along the land/sea boundary,
- Apply a cap to the coupled Charnock values larger than 0.32, to better represent strong wind conditions.

The ST4 source term parameterization scheme (Ardhuin et al., 2010) is used in all RCS-IND1 wave configurations. The linear input source term parameterization of Cavaleri and Malanotte-Rizzoli (1981) is applied to improve initial wave growth behaviour. Non-linear wave-wave interactions are parameterized following the Discrete Interaction Approximation
(Hasselmann et al., 1985). Bottom friction is taken into account with the JONSWAP formulation (Hasselmann et al., 1973), and depth-induced breaking using the Battjes and Janssen (1978) approach. Wind and current forcing are linearly interpolated in time and space at the coarsest grid scale (4.4 km), and the wind speed forcing is corrected relative to the ocean current velocity.

When run in uncoupled modes, the wave model can be forced (see Table 6) with ocean currents (obtained from a regional ocean model, such as the ocean-only component of RCS-IND1) and atmospheric wind, or just atmospheric wind



(*WV_OCN_FORCING = true/false*, respectively). The atmospheric wind forcing can be provided via global NWP model data (*UM_FORCING = flx-global*) or regional km-scale model simulations (*UM_FORCING = flx-high*).

The wave component in all simulations presented in this paper is initialized from a restart file generated by first running the RCS-IND1 wave-only configuration from rest for a 5-day period prior to a required case study initial time. In these spin-up simulations, wind forcing is provided by the operational global UM forecast archive, running at approximately 17 km resolution at tropical latitudes at the time of the case studies described in this paper. Spectral boundary conditions were provided from archived operational global wave model output.

## 335 3 RCS-IND1 performance and the impact of coupling: TC Titli and TC Fani case study assessment

The sensitivity of TC simulation to model coupling using the RCS-IND1 configuration has been assessed by Saxby et al. (2021) for six TCs that developed in the Bay of Bengal between 2016 and 2019. They consider RCS-IND1 performance across a range of model lead times and focus on the impact of coupling from analysis of atmosphere-only uncoupled and atmosphere-ocean coupled simulations. In this paper, the full flexibility of the RCS-IND1 framework is demonstrated for
two of the cases assessed by Saxby et al. (2021), namely Cyclone Fani (April 2019; pre-monsoon, e.g., Routray et al., 2020; Zhao et al., 2020; Singh et al., 2021) and Cyclone Titli (October 2018; post-monsoon, e.g., Mahala et al., 2019; Maneesha et al., 2021). Here, the performance of a broader range of coupled simulation approaches and configuration options is considered for a single initialisation time to examine the potential diversity of results.

### 3.1 RCS-IND1 sensitivity experiments

The different simulation approaches demonstrated in this paper for 7-day simulations of Titli (initialised at 0000 UTC on 8 October 2018) and Fani (initialised at 0000 UTC on 28 April 2019) cases are summarised in Table 7 (see also Fig. 2). Saxby et al. (2021) considered atmosphere-only and atmosphere-ocean coupled simulations for a variety of initialisation times, but the focus on the initialisation and run duration discussed here represent a balance of capturing much of each storm's lifecycle and subsequent impacts. Titli formed on 8 October 2018, reaching landfall in Andhra Pradesh around 00UTC on 11 October
(i.e. 3 days into simulations presented here), and dissipated on 12 October. Fani was a long-lived storm, forming on 26 April 2019, two days prior to the initialisation time assessed here, but did not reach landfall in Odisha until 0230 UTC on 3 May 2019 (i.e. over 5 days into simulations), and dissipated on 5 May 2019, 9 days after first forming.

Two types of atmosphere-only control simulations are considered using the following naming conventions:

• *ATMfix*: initial condition OSTIA SST surface boundary is kept constant throughout the run of RCS-IND1a configuration,





- ***ATM***: SST field is updated daily with the OSTIA product generated by 00 UTC on each day through the simulation period, to reflect the data that would have been available at that (re)-initialisation time if running in near-real time.

Three approaches to representing feedbacks between atmosphere and ocean are then considered:

- ***KPP***: the RCS-IND1ak coupled configuration is used, where the UM atmosphere component is coupled to the 1D mixed layer KPP multi-column parameterisation (Section 2.5) each hour through the simulation.
- ***AO***: the RCS-IND1ao configuration is used with two-way coupling between the atmosphere and the 3D NEMO ocean model component (Section 2.4).
- ***AOW***: the fully coupled RCS-IND1aow configuration is run with hourly two-way coupling between atmosphere,
ocean and WAVEWATCH III wave model components (Section 2.2; Table 3).

All simulations presented are run for 7 days from a common initial atmosphere condition, with differences of ocean and wave initial conditions described for experiment in Section 2. The lateral boundary forcing at the domain edges is common across all experiments, using the same global-scale boundary conditions relevant to atmosphere, ocean and wave components.


Given the focus of application in this paper on simulation of TCs, the relative sensitivity of simulations to whether the dissipation of turbulent kinetic energy is included in the UM boundary layer parameterisation is also considered. Turbulent motions are ultimately dissipated as heat, which can result in a significant contribution to the energy budget from frictional heating particularly at stronger wind speeds (e.g., Kilroy et al., 2017). This term can be computed or omitted in the UM
through setting a parameter option (*fric_heating*), which is typically enabled in global-scale model configurations running with grid spacing of order 10 km or coarser. Frictional heating is however typically disabled in higher resolution regional model configurations (e.g., Bush et al., 2020) as a pragmatic approach to limiting the tendency to over-intensify strong cyclones when running convective-scale atmosphere simulations. The move towards coupled predictions and more explicit representation of air-sea interactions requires this to be re-examined. All simulations listed in Table 7 have therefore been
performed without (*fric_heating=0*) and with (*fric_heating=1*) the frictional heating term added to the computation of sub-grid turbulent kinetic energy budget. In the current UM formulation, the additional heating is applied uniformly in the vertical over the boundary layer. Simulations with frictional heating enabled have ***_FH*** added to the respective experiment identifier when referred to in the text (e.g., ***ATMfix_FH*** for fixed SST atmosphere-only simulation with frictional heating enabled).

**3.2 Impact of coupling on representation of SST**

The variety of approaches to representing SST within the RCS framework is illustrated in Figs. 3 and 4 for Titli and Fani cases respectively. The *ATMfix* simulations exemplify the assumption imposed in most operational regional atmosphere forecasting systems, whereby the initial condition OSTIA (representative of satellite observed foundation SST typically two days prior to initialisation time) is persisted throughout the simulation. Note this is typically only applied over a forecast





duration of a few days, for example with the NCUM-R regional forecasts currently run to 76 h (Routray et al., 2020), while

the simulations used in the current case studies extend for twice as long. While only feasible for 'hindcast' case study

assessments rather than operational forecasting, the *ATM* simulations in Figs. 3(d) and 4(d) show the Bay of Bengal sub-

region mean OSTIA SST becomes around 0.5 K cooler when applying daily updating OSTIA SST over the 7-day period of

the Titli and Fani cases. In contrast, the capability for ocean model components in *KPP*, *AO* and *AOW* configurations to

simulate both cyclone-induced cooling over the duration of the cyclone evolution and diurnal heating effects of order 0.5 K

is evident.

One of the limitations of the current experimental design of RCS-IND1 is that a free-running ocean-only simulation has been

used to spin-up the small-scale dynamics in the 3D ocean model component used in *AO* and *AOW* coupled runs. For the Titli

case, the ocean initialisation (Fig. 3(c)) is on average 0.6 K cooler than OSTIA, noting this is larger than the magnitude of

mean observed cyclone-induced cooling during the event. Figure 3(c) shows this cooling is broadly distributed across the

Bay of Bengal, though as much as 2 K cooler towards the eastern side. The *KPP* simulation for Titli is initialised on average

about 0.2 K cooler than OSTIA, but with more varied spatial distribution of initial condition differences. The mean initial

condition for the Fani case is much more similar between all experiments, albeit with *KPP* initialisation slightly warmer than

OSTIA in central and northern Bay of Bengal. When interpreting differences in the atmospheric response between

experiments, it should be noted that not all experiments could be initialised from a common initial SST. Note also that while

the SST imposed on the atmosphere in *ATMfix* and *ATM* are an observation-based foundation SST, representative of order

10 m below the ocean surface, the coupled system involves exchanging the top ocean model level temperature, typically

within 1 m of the surface (e.g., Mahmood et al., 2021).


The diurnal cycle heating effect on SST evolution can be seen in Fig. 3(d) and Fig. 4(d), with *AOW* ocean temperatures

around 0.4K warmer during day than at night in periods when the influence of cyclones was less prevalent, such as after

passage of Titli (e.g., 14 October) and before passage of Fani (e.g., 28 April 2019). Wave coupling leads to slightly reduced

magnitude of diurnal warming than in *AO* for both cases, consistent with wave-enhanced mixing in *AOW*.


The magnitude and spatial distribution of cyclone-induced cooling is shown in Fig. 3(e)-(h) and 4(e)-(h). Based on OSTIA

data, Titli led to a decrease of 0.6 K in foundation SST, spread relatively evenly across the northern Bay of Bengal with the

largest cooling over the whole period of 2.4 K. For *KPP_FH*, the maximum temperature decrease is 3.0 K, although the

imprint of the passage of Titli can be more clearly seen in Fig. 3(f) than the OSTIA-based observations in *ATM_FH* (Fig.

3(d)). Larger but more focussed cooling of up to 3.8 K (*AO_FH*; Fig. 3(g)) and 4.3 K (*AOW_FH*; Fig. 3(h)) are evident when

coupling to a 3D NEMO ocean component. Similar features can be seen for the Fani case. Figure 4(d) demonstrates that the

OSTIA data available on 4 May 2019 (representative of satellite-observed ocean temperatures on 2 May 2019), on average

0.5 K cooler than on 28 April, does not yet represent the full extent of cyclone-induced cooling across the region. The largest





OSTIA temperature reduction in the central Bay of Bengal during the period is 2.5 K (*ATM*; Fig. 4(e)), in contrast to more
focussed and stronger cooling in coupled simulations of up to 4.8 K (*KPP_FH*; Fig. 4(f)), 5.9 K (*AO*; Fig. 4(g)) and 6.5 K
(*AOW*; Fig 4(h)). The more intense cyclone-induced cooling in AO and AOW than in KPP is consistent with the absence of
upwelling using a 1D approach (Yablonsky and Ginis, 2009).

The different approaches to representing SST are compared to in-situ ocean buoy data from 3 illustrative sites located in the
central Bay of Bengal (see Fig. 1) in Fig. 3(i)-(k) and Fig. 4(i)-(k). This analysis is complicated by the different ocean
initialisation approaches required across experiments, positional differences in cyclone track and intensity evolution, along
with relatively infrequent and coarse numerical precision of reported ocean observations. There are also discrepancies
between ocean buoys sampling within order 1 m of the surface, NEMO ocean temperatures representing the upper ocean
model layer, and OSTIA representing a foundation SST. However, it is evident that *AO* and *AOW* simulations capture the
scale of ocean cooling relatively well for the Titli case, though with cooling tending to initiate up to a day earlier than
observed. Accounting for the *KPP* simulations being slightly warmer than observed at initialisation, the magnitude of
cooling is stronger than observed during Titli at buoy locations 23093 and 23091, but with *KPP* temperatures remaining too
warm throughout the simulation at 23459 further south. It would be interesting to explore the sensitivity of Bay of Bengal
SST to the representation of lateral advection, for example by running the NEMO ocean component without tides (e.g.,
Arnold et al., 2021), to better understand their influence on the *KPP* results.

For the Fani case, given that the initial condition OSTIA data are up to 0.5 K cooler than buoy observations, the persisted
(*ATMfix*) and daily updating (*ATM*) SST assumptions in fact provide a reasonable 7-day approximation to observed
temperatures in the northern Bay of Bengal (e.g., 7-day mean bias of -0.22 K at 23093 and 0.03 K at 23091 for *ATMfix* SST,
relative to -0.70 K and -0.33 K for *AOW_FH* at those buoys). The *KPP* runs are initialised with SST in good agreement with
the observed buoys and provide a good representation of both the diurnal cycle ahead of Fani passing and of the observed
cyclone-induced cooling later during the period at all three buoy locations. This leads to smallest root mean squared errors of
all experiments for the *KPP* run (i.e., without frictional heating) of 0.25 K, 0.34 K and 0.18 K at buoys 23459, 23093 and
23091 respectively. Initial condition errors for *AO* and *AOW* are preserved through the simulations relative to observed SST,
and there is some evidence that *AO* and *AOW* cool too much as the cyclone passes (e.g., 00 UTC on 2 May at 23459 and 00
UTC on 4 May at 23091). If removing initial condition offsets however (not shown), *AO* has smallest mean bias and root
mean squared errors (RMSE) of all experiments relative to observations for buoy location 23091 (RMSE of 0.16 K) and
23093 (RMSE of 0.23 K), whereas *KPP* has slightly lower RMSE at 23459 (RMSE of 0.44 K).

The SST timeseries in Figs. 3 and 4 also highlight some sensitivity in *KPP*, *AO* and *AOW* simulations to the representation
of frictional heating in the coupled atmosphere component. Small differences in SST evolve after about 3 (Titli) or 4 (Fani)
days between equivalent simulations with or without frictional heating, with the introduction of frictional heating leading to





stronger induced ocean cooling (due to more intense storm development), and SST of order 0.2 K cooler than for runs without this process enabled. The fit relative to observed SST, after removing initial condition offsets, is similar but slightly

degraded in coupled simulations using frictional heating for the locations plotted in Figs. 3 and 4. It can also be concluded that the influence of frictional heating is of secondary importance relative to the choice of coupling approach for SST evolution over the 7-day period for both cases.

### 3.3 Tropical cyclone structure, track and intensity

Cyclone tracks have been diagnosed from mean sea level pressure (MSLP) and vorticity model diagnostics for each

experiment conducted using the method described by Heming (2017). Simulated maximum wind speed and diagnosed tracks for the Titli case are compared in Fig. 5 (for experiments with frictional heating), alongside an illustration of the observed cyclone structure as it neared landfall from Meteosat satellite imagery at 00 UTC on 10 October 2018 and a multi-agency observed track as diagnosed and shared via the Global Telecommunication System (GTS) in near-real time during each event, based on bulletins from Regional Specialised Met Centres (RSMCs), tropical cyclone warning centres, and the Joint

Typhoon Warning Center. Contours of outgoing longwave radiation at the equivalent time are plotted to illustrate the simulated cyclone structure. Corresponding results for simulations without frictional heating are provided for reference in Supplementary Material Figure S1. Results for Cyclone Fani are presented in Fig. 6 (see also Supplementary Material Figure S2), where the observed cyclone structure is shown by brightness temperatures from the INSAT satellite at 06 UTC on 2 May 2019. Results from both cases provide qualitative evidence that the RCS-IND1 configuration can be used to simulate

intense storm development over the Bay of Bengal with realistic cloud structures and peak simulated winds aligning with the diagnosed model track that are generally well aligned to observed tracks.

Figures 7 and 8 provide a more quantitative comparison of the diagnosed track and intensity for all experiments considered for the Titli and Fani cases respectively. Summary metrics are provided in Table 8. An important result from Figures 7 and 8

is evidence that while coupling tends to drive the same differences between experiments whether run with or without frictional heating, including the frictional heating contribution to the boundary layer energy budget can have as large an impact on cyclone characteristics as the introduction of model coupling. The relative impact of coupling on results with and without frictional heating is therefore considered within the same discussion below.

While Fani was considerably longer lived and more intense than Titli, the sensitivity of cyclone intensity to coupling is similar for both cases. The lack of air-sea interaction in *ATMfix* (*ATMfix_FH*) results in deeper cyclones, by 33 hPa (31 hPa) for Titli and 36 hPa (30 hPa) for Fani, compared to the fully coupled *AOW* (*AOW_FH*) simulations. This is consistent with the well-established result that the representation of surface cooling feedbacks in coupled simulations tends to limit cyclone intensification (e.g., Vincent et al., 2012; Feng et al., 2019; Vellinga et al., 2021; Saxby et al., 2021).





### 3.3.1 Cyclone Titli

For Titli, the *ATMfix* and *ATM* uncoupled and *KPP* coupled simulations are found to over-deepen relative to the observed intensification, while the *AO* and *AOW* coupled results with a 3D ocean model component are considerably closer to the observed minimum pressure. This conclusion is common to simulations with and without frictional heating. The over-deepening is consistent with relatively warmer initial condition SST in the *KPP* and uncoupled simulations in the cyclone genesis region (Fig. 3(b), Fig. 3(i)), and the simulated cyclone being too intense may have contributed to the excessive cooling of *KPP* in the northern Bay of Bengal relative to observations (Fig. 3(k)). All cyclone simulations deepen more quickly than observed, although *AO* and *AOW* intensify later than the uncoupled and *KPP* simulations for this case. The addition of wave coupling in *AOW* results in slightly earlier intensification than *AO*, particularly with frictional heating (Fig. 7(e)).

Smallest cyclone track errors are evident for the *ATMfix(_FH)* experiments (Fig. 7(a),(d)), with tracks deviating westward in *ATM(_FH)* and all coupled experiments. This westward trajectory is particularly pronounced for the *KPP(_FH)* coupled simulations (Fig. 5(d)). Katsube and Inatsu (2016) illustrated a tendency for TCs in the north-west Pacific to recurve faster over relatively warmer oceans for storms in the north-west Pacific, and thereby propagate relatively rightward of a simulation with intermediate SSTs. Over relatively cooler oceans, they found slower re-curvature and leftward propagation. The RCS framework provides the capability to perform such sensitivity experiments for these cases, and it would be of interest to examine if similar processes account for track deviation in the Bay of Bengal, although beyond the scope of the current study. Rather, it can at least be noted that the relatively westward propagation is consistent with more slowly intensifying cyclones in coupled cases over a relatively cooler ocean. The reasons for *KPP* deviating so far to the west for the Titli case, further westward than *AO* and *AOW*, resulting in largest track errors, is however unclear. For *KPP*, *AO* and *AOW*, the track trajectory is improved with representation of frictional heating (i.e., westward tendency reduced), consistent with relatively quicker storm intensification and deeper storms developing.

### 3.3.2 Cyclone Fani

By contrast, the coupled simulations for Fani, which deepened considerably to an observed minimum MSLP of 917 hPa, are considerably too weak (minimum simulated central pressure in AOW of 959 hPa) and, unlike for Titli, the *KPP* results are now more similar to the *AO* and *AOW* coupled simulations. It is notable that none of the simulations captured the initial northward propagation and relative delay to intensification of Fani's evolution, with all simulated storms deepening within the first day of simulation and thereby veering westward too early. This suggests errors in the common initial conditions inherited from the global model analysis. Further analysis (not shown) indicated that the initial stages of the Fani simulations may have been degraded by distortion of the cyclonic vortex within the global operational analysis used in the initialisation,





suggesting that RCS-IND1 results may improve with implementing a vortex initialization scheme for experiments specially aimed at simulating tropical cyclones (e.g., Liu et al., 2020).

All simulated storms turned northward from around 3 days into the simulation, when the uncoupled simulations further deepened and track errors were much reduced (Fig. 8(a),(d)). Over-deepening over a relatively warm ocean SST is associated with a rightward propagating cyclone for *ATMfix(_FH)* and *ATM(_FH)*, such that *ATMfix(_FH)* tracks to the east of the observed track. This is more pronounced when frictional heating is applied (*ATMfix_FH*). In contrast, each of coupled simulations *KPP(_FH)*, *AO(_FH)* and *AOW(_FH)* track along the observed path from 30 April 2019, with *AOW(_FH)* having slightly improved track relative to *AO(_FH)*. After landfall early on 3 May 2019, the diagnosed tracks in coupled 530 simulations progress too far north, consistent with relatively slower deintensification. Track errors are reduced when frictional heating is included in all coupled simulations, consistent with more intense storms developing.

### 3.3.3 Summary impact of frictional heating

The approach of not including the heating term in uncoupled regional UM configurations to date as a pragmatic means to improving model results seems to be supported in results for both Titli and Fani cases with *ATMfix_FH* and *ATM_FH* over-535 deepening and having increased MSLP and track errors relative to the equivalent *ATMfix* and *ATM* simulations without frictional heating. Track errors are more impacted for Fani than Titli, for which *ATM_FH* considerably over-deepens from 1 May to a minimum central MSLP as low as 907 hPa. In contrast, addition of frictional heating improves track errors for *AO* and *AOW* coupled simulations for both Titli and Fani, and there seems to be some improvement to the timing of the dissipation phase. While more intense storms are simulated for Fani with *AO_FH* and *AOW_FH* than for the equivalent runs 540 without frictional heating, its impact is insufficient to deepen as much as observed or uncoupled simulations. There is also some indication that the relative impact of coupling on simulations may be slightly reduced with frictional heating (i.e., range of pressure and wind speeds smaller between experiments). These results lead to the recommendation that while it continues to be pragmatic to disable frictional heating when running the UM in uncoupled modes, coupled results can be improved when frictional heating is active. In summary, by representing coupled feedbacks it appears possible and desirable 545 to include an additional term in the energy budget of regional simulations and thereby provide a fuller representation of the physics of tropical systems.

### 3.4 Impact of coupling on wind speed

In general, the comparison of track-diagnosed wind speed relative to those indicated in the real-time GTS bulletins in Figs. 7 and 8 show that all simulations under-predicted peak wind speeds, particularly for the Fani case. The over-deepening of 550 uncoupled simulations for both cases results in closer agreement to the observation-based peak wind, while frictional heating also leads to deeper and thereby stronger winds. Saxby et al. (2021; see their Figure 5) illustrated from their analysis of more simulated cases and a range of initialisation times that the wind-pressure relationship for km-scale UM simulations has





generally too low winds for given MSLP relative to observations above around 35 ms$^{-1}$, with those errors being reduced but not eliminated with coupling. The current case study results highlight that with frictional heating, the UM can generate
intense storms with insufficient maximum wind speeds (i.e., deepest MSLP of 907 hPa with maximum wind speed 104 kn for *ATM_FH* in Table 8), while in the equivalent coupled simulations, cyclones do not deepen sufficiently albeit with relatively stronger wind speeds for given MSLP (e.g., *AOW_FH* deepened to 948 hPa with maximum wind speed 98 kn).

These results are supported by a comparison of simulated wind speed at 10 m above the surface with in-situ observations
near landfall on the Indian coast (Gopalpur; Figure 1) and at two ocean buoy locations in the Bay of Bengal, shown for Titli and Fani cases in Figure 9. As discussed in Section 3.2, quantitative comparison with observations is challenging due to the different cyclone tracks in each simulation, and potentially substantial observation errors both during extreme conditions and above the ocean. Some caution might therefore be applied to the apparent tendency for simulations to have stronger wind speeds than observed by ocean buoys during both cases, although is it clear that stronger winds are simulated with fixed and
daily updating SST than for any of the coupled simulations, which tend to have improved bias and RMSE statistics. The too rapid deintensification of Titli in uncoupled simulations is also evident in comparison with observations at 23091 (Fig. 9(c)), where the beneficial impact of frictional heating can be seen by the final simulation day.

Consistent with along-track results for Titli (Fig. 7(c),(d)), comparisons at Gopalpur (Fig. 9(a)) show that only peak
simulated wind speed with *ATMfix* of 21 ms$^{-1}$ start to approach the observed maximum wind speed of around 30 ms$^{-1}$. The timing of maximum wind speeds matches observations well, however. *AO_FH* provides the best match to observed peak winds at Gopalpur for the Fani case (Fig. 9(d)), although the timing is slightly delayed relative to observations. These results also clearly show peak winds too early in uncoupled simulations relative to observations, noting that peak wind speeds from the uncoupled simulations are relatively weaker at this location given that the simulated storm tracked further eastward than
observed (Fig. 6(b),(c)). The impact of wave coupling on wind speeds is relatively small during both cases. Some improvements to the timing and magnitude of maximum winds with frictional heating is evident for *KPP_FH*, *AO_FH* and *AOW_FH* simulations relative to equivalent runs without frictional heating. However, in comparison with the clear impact on cyclone track and intensity, the sensitivity of cyclone winds away from the cyclone track to frictional heating is relatively small for these cases.

Developments to improve the wind speed characteristics of the RCS-IND1 configuration are in progress, and their impact will need to be evaluated in future studies. A key consideration is the representation of surface drag at high wind speeds (more than 30 ms$^{-1}$). Different approaches to change the UM drag parameterisation under investigation were discussed recently by Gentile et al. (2021a) in the context of km-scale coupled UM simulations of extratropical cyclones. This includes
testing the impact of moving to the COARE 4.0 parameterisation at lower wind speeds, with a cap and reduction in drag coefficient at higher wind speeds. In the RAL1 physics configuration used in the current study, the drag coefficient is





assumed to increase linearly as a function of wind speed, implying that winds are excessively dampened at higher wind speeds. This is known to be unrealistic, with Donelan (2018) for example arguing that a reduction in the drag coefficient above 30 ms$^{-1}$ was critical to representing rapid intensification. It will therefore be important to re-examine these and other cyclone cases using revised RAL physics definitions. For example, Baki et al. (2021) found simulation of TCs in Bay of Bengal could be improved by up to 16% for wind speed using optimal parameters of the WRF model based on sensitivity analysis of a range of physics parameters.

### 3.5 Impact of coupling on precipitation

The impact of coupling on accumulated precipitation is illustrated for Titli and Fani cases in Fig. 10 and Fig. 11 respectively and a more quantitative comparison of the domain-accumulated precipitation shown in Fig. 12 and Fig. 13. Results are compared with the NASA GPM (Global Precipitation Measurement; Hou et al., 2014) IMERG observations, with all precipitation data interpolated to the GPM resolution of 0.1° prior to analysis. The influence of model spin-up from global-scale atmosphere initialisation can be seen during the first day of each simulation (Fig. 12(a)(d) and Fig. 13(a)(d)) and thereby the first day is omitted from the following analysis. Figures 10 and 11 demonstrate relatively good simulation of the spatial extent of precipitation across the Bay of Bengal associated with both cyclones and their subsequent eastward passage following landfall. All simulations tend to have too little light precipitation, which is a common feature of convective-scale UM simulations with RAL1-T configuration (Bush et al., 2020). This is illustrated by relatively fewer accumulations of less than 100 mm in all simulations than observed by GPM in Fig. 12(c)(f) and Fig. 13 (c)(f). There is however better agreement with GPM for the relative frequency of higher accumulated precipitation totals.

The over-intensification of uncoupled simulations of Titli is evident in Fig. 12 with *ATMfix(_FH)* accumulated precipitation consistently higher than observed after 11 October, contributing to a net over-prediction of accumulated precipitation of 18% (21%) for *ATMfix* (*ATMfix_FH*) and 11% (12%) for both *ATM* (*ATM_FH*) simulations. Coupled simulations have a net deficit of accumulated precipitation during the first half of the Titli case study relative to GPM, but over the 6-day period KPP has slightly higher accumulated precipitation (3% higher than GPM for *KPP* and 8% for *KPP_FH*) while *AO(_FH)* and *AOW(_FH)* are well matched (biases of *AO*: -1%, *AO_FH*: 0%; *AOW*: -2%; *AOW_FH*: 1%). For the Fani case (Fig. 13), all simulations miss the peak in observed precipitation on 30 April 2019, perhaps associated with the lack of initial northward cyclone propagation. *ATMfix* and *ATM* simulations then have relatively good estimates of Bay of Bengal regional accumulation (6-day accumulation bias of 1.5% and 0.5% respectively), but with higher accumulations when frictional heating was applied, consistent with a more intense simulated cyclone (biases of 6% for *ATMfix_FH* and 2% for *ATM_FH*). For this case, the *KPP*, *AO* and *AOW* results tend to underpredict accumulated precipitation (by 9% for *KPP* and 12% for *AO* and *AOW*), particularly after landfall early on 3 May 2019, with enhanced precipitation and a slightly improved agreement relative to GPM with frictional heating (bias of -6% for *KPP_FH* and -10% for both *AO_FH* and *AOW_FH*).




In common with the wind speed results, it will be valuable to re-examine the impact of using revised RAL configurations on the RCS-IND1 precipitation characteristics. For example, development of a new bimodal diagnostic cloud fraction (Weverberg et al., 2021) and cloud microphysics (e.g., Hill et al., 2015) parameterizations in RAL offer pathways towards improving the frequency distribution of simulated precipitation. Improving the representation of precipitation in RCS-IND1

is a key priority in the context of coupled prediction given the opportunity to further assess and develop the land surface model component to enable a more integrated approach to simulating the terrestrial water cycle (e.g., Lewis and Dadson, 2021). This is of particular importance in the Bay of Bengal given potential feedbacks through the ocean state (e.g., Krishnamohan et al., 2019).

### 3.6 Computational resources

Table 9 provides a summary of the computational resources required to run different RCS-IND1 configurations of the RCS modelling framework. Simulations discussed in this paper were conducted on the Met Office Cray XC40. Reported values indicate that the RCS provides a suitable tool for running research configurations within a practical time limit, with configurations typically completing a day simulation within order 20 minutes runtime. Run times for comparable simulations run on the NCMRWF Cray XC40 are also listed, with the RCS having been successfully ported to that machine to enable

ongoing collaboration and motivate new simulation experiments. Considerable opportunities for system optimisation are thought to exist in both the regional model components and coupling interfaces, which will be implemented in future updates. For example, updating the wave model component from WAVEWATCH III vn4.18 to vn7.0 is anticipated to enable coupling to be performed independently between each model processor, rather than coupling via a single processor as required at present.

**4 Discussion and ongoing development**

A new implementation of a flexible regional coupled modelling framework focussed on the Indian region has been described. The primary motivation for this development is to provide underpinning capability for research into the sensitivity of hazardous weather and its impacts to how interactions are represented within simulations of the environmental system. This research may ultimately lead to improved operational predictions and services delivered through the Indian Ministry of

Earth Sciences. Given the high population density, particularly in coastal regions, and prevalence of natural hazards linked to the Indian monsoon progression, these research questions and operational impacts are of critical importance.

This paper documents the scientific and technical basis of the RCS-IND1 implementation, with aspects of its flexibility to support a range of experimental designs highlighted to motivate a breadth of future research activities using these

capabilities. Results have been presented to demonstrate the sensitivity of simulations of cyclone Titli and Fani with a

variety of approaches to the representation of the ocean, including uncoupled atmosphere simulations with fixed SST (*ATMfix*), daily updating OSTIA (*ATM*), a simplified coupled system with the ocean represented by a 1D mixed layer parameterisation (*KPP*), and coupling to a 3D ocean model (*AO*) or coupling to both ocean and wave models with two-way interactions between all components (*AOW*).


The relative influence of frictional heating in the UM boundary layer formulation has also been examined. This study confirms that the uncoupled simulations still tend to be optimised without frictional heating. While the sensitivity to coupling is consistent with and without frictional heating, results show that coupling effectively enables this term to be included in a convective-scale simulation. Although Fani was a stronger storm than Titli, and the effect of frictional heating might be

expected to be more significant for more intense storms, these results show a broadly similar difference between results with and without frictional heating for each case. A broader study of the sensitivity of coupled results to frictional heating, in particular to assess the sensitivity of runs with earlier initialisation times would be of interest to assess its impact during initial cyclogenesis (e.g. Kilroy et al., 2017).

All simulations demonstrate some long-standing model biases that are not substantially corrected through model coupling, such as a tendency for winds to be too light for a given MSLP, and insufficient light rain. While the introduction of air-sea interactions through coupling markedly improves the intensification of Titli to be closer to observations, the reduced intensification leads to poorer simulation of minimum pressure but improved cyclone track prediction for Fani. These results are consistent with the analysis of Saxby et al., (2021), who provide a review of RCS-IND1 performance for *ATM* and *AO*

configuration across a broader range of cyclone cases.

For the two TC cases discussed in this paper, coupling with the waves shows smaller impact than coupling with the ocean. This contrasts with the sensitivity found for extra-tropical cyclones (Gentile et al. 2021a), potentially as the wave feedback on drag saturates for the higher wind speeds found in TCs.


For these cases, a 1D mixed layer scheme shows similar first-order SST cooling and feedback on the atmosphere as for coupling to a full 3D ocean model. Nevertheless, the 3D ocean generally shows stronger localised cooling than the 1D mixed layer ocean. This is consistent with shear-induced mixing of the upper ocean being the main cooling mechanism, with 3D ocean upwelling playing a secondary role (Yablonsky and Ginis, 2009). As discussed by Singh et al. (2021) for the Fani

case, effective incorporation of ocean initial conditions (surface and sub-surface) is vital for effective representation of cyclone genesis and intensification, and thereby improving these aspects in the RCS framework remains a priority.

Further scientific and technical development of the RCS-IND1 configuration is planned. Key research priorities include:





- Analysis of ocean and wave performance of RCS-IND1, for example associated with the ocean response to cyclone evolution,

- Improving the initialisation of components, in particular the regional ocean, for example based on initialisation from equivalent uncoupled regional analyses or through developing regional weakly coupled data assimilation,

- Reviewing and improving the conservation of the surface momentum budget across the atmosphere-wave-ocean interface and its treatment between the 3 component models,

- Understanding the sensitivity of air-sea interaction and optimising results to the choice of coupling frequency,

- Demonstrating and assessing application of RCS-IND1 for concurrent multi-hazard prediction, such as wind-tide-surge interactions and coastal flooding,

- Understanding the impact of choice of lateral boundary conditions on system performance, for example to establish the sensitivity to use of coupled or uncoupled global model boundaries,

- Assessment of the impact of convective-scale precipitation on the land surface, and thereby representation of the terrestrial water cycle, river flows and discharge to the ocean,

- Examination of sensitivity to coupling for a broader range of meteorological cases, for example of monsoon depressions, or simulations over longer timescales.

Key technical developments to the RCS, that will be tested for the India-focussed domain, will include:

- Addition of capability to run coupled experiments in ensemble mode, to explore the relative sensitivity of coupled results to the model spread introduced through initial condition and stochastic perturbations (e.g. Gentile et al., 2021b).

- Improve the representation of climatological freshwater inflow to the ocean component, before later adding simulation of river flow and surface inundation within the JULES land surface model, thereby enabling a more integrated treatment of the hydrological cycle between atmosphere, land and ocean components (e.g. Pandey et al., 2021),

- Upgrade atmosphere, ocean and wave model codes and scientific configurations to more recently available versions and examine the impact on system scientific and computational performance,

- Improving flexibility of pre-processing and domain setup workflows within the modelling framework, to further simplify the process of establishing new regional coupled domains to support further research.

It should also be noted that different coupled and uncoupled implementations of RCS-IND1 have been successfully run over longer periods of up to a month as part of its development. This continues to be an exciting time in the development and application of coupled tools to better understand the role of environmental interactions at regional scales. The RCS modelling framework provides the flexibility required to better understand the role of different feedbacks and processes



within the system, with the prospect that this will lead to improved operational services and information to better protect lives and livelihoods in the years ahead.

**Code availability**

Due to intellectual property right restrictions, neither the source code nor documentation papers for the Met Office Unified Model or JULES can be provided directly through open-source repositories. All model codes used within the RCS-IND1 configuration are however accessible to registered researchers, and links to the relevant code licences and registration pages are provided for each modelling system below. All code used can also be made available to the Editor and reviewers for review. The supplement to this paper includes a set of namelist parameters and their settings that define the atmosphere,

land, ocean and wave configurations in RCS-IND1 simulations. All codes used to generate the analysis discussed in Section 3 is available to registered collaborators at https://code.metoffice.gov.uk/trac/utils/browser/ukeputils/trunk/ukep_plot (last access: 27 October 2021).

**Obtaining the Unified Model**

The Unified Model (UM) is available for use under licence. A number of research organizations and national meteorological services use the UM in collaboration with the Met Office to undertake basic atmospheric process research, produce forecasts, develop the UM code and build and evaluate models. For further information on how to apply for a licence see https://www.metoffice.gov.uk/research/approach/modelling-systems/unified-model/index (last access: 27 October 2021). The UM vn11.1 trunk code and associated modifications for RCS-IND1 are available to registered researchers via a shared

UM code repository, which can be accessed via https://code.metoffice.gov.uk/trac/um/wiki (last access: 27 October 2021). Details of the separate code branches with modifications for RCS-IND1 are documented in the Supplementary Material. A copy of the merged UM code used for RCS-IND1 is provided at https://code.metoffice.gov.uk/trac/utils/browser/ukeputils/trunk/gmd-2021/ind1/um (last access: 5 January 2022) to support collaboration.

**Obtaining JULES**

JULES is available under licence free of charge. For further information on how to gain permission to use JULES for research purposes see http://jules.jchmr.org (last access: 27 October 2021). The JULES vn5.2 trunk code and associated modifications for RCS-IND1 are freely available on the JULES code repository, which can be accessed via https://code.metoffice.gov.uk/trac/jules/wiki (last access: 27 October 2021). Details of the separate code branches with

modifications for RCS-IND1 are documented in the Supplement. A copy of the merged JULES code used for RCS-IND1 is provided for reference and to support collaboration at https://code.metoffice.gov.uk/trac/utils/browser/ukeputils/trunk/gmd-2021/ind1/jules (last access: 5 January 2022).

**Obtaining NEMO**



The model code for NEMO vn4.1 is available from the NEMO website (https://www.nemo-ocean.eu/, last access: 27
October 2021). After registration the Fortran code is readily available to researchers. A copy of merged code branches at
https://code.metoffice.gov.uk/trac/utils/browser/ukeputils/trunk/gmd-2021/ind1/nemo (last access: 5 January 2022) contains
modifications to the NEMO vn4.0.1 trunk applied for RCS-IND1. A list of the NEMO compilation keys applied on building
the merged NEMO code is provided in the Supplementary Material. Also provided are details of the separate code branches
with modifications for RCS-IND1.

**Obtaining KPP**

The KPP code is available via the PUMA website (http://cms.ncas.ac.uk/wiki/PumaService last access: 27 October 2021)
after contacting the Computation Modelling Services of the National Centre for Atmospheric Science. See Supplementary
Materials for further detail. For reference and to support collaboration, a copy of the KPP branch used in this study is
provided at https://code.metoffice.gov.uk/trac/utils/browser/ukeputils/trunk/gmd-2021/ind1/kpp (last access: 5 January
760 2022).

**Obtaining WAVEWATCH III**

The WAVEWATCH III® code base is distributed by NOAA National Weather Service Environmental Modeling Center
under an open-source-style licence via https://polar.ncep.noaa.gov/waves/wavewatch/wavewatch.shtml (last access: 27
October 2021). Interested readers wishing to access the code are requested to register to obtain a licence via
https://polar.ncep.noaa.gov/waves/wavewatch/license.shtml (last access: 27 October 2021). The model is subject to
continuous development, with new releases generally becoming available to those interested and committed to basic model
development, subject to agreement. Model codes used in the RCS-IND1 system are maintained under configuration
management via a mirror repository hosted at the Met Office. A copy of which is provided to researchers for collaboration
on request at https://code.metoffice.gov.uk/trac/utils/browser/ukeputils/trunk/gmd-2021/ind1/ww3 (last access: 5 January
2022), given prior approval to access WAVEWATCH III from NOAA. The Supplement provides a list of the
WAVEWATCH III compilation switches applied on building the wave model code.

**Obtaining OASIS3-MCT**

OASIS3-MCT vn2.0 is disseminated to registered users as free software from https://oasis.cerfacs.fr/en/ (last access: 27
October 2021).

**Obtaining Rose**

Case study simulations and configuration control namelists were enabled using the Rose suite control utilities. Further
information is provided at http://metomi.github.io/rose/doc/html/index.html (last access: 27 October 2021), including
documentation and installation instructions.

**Obtaining FCM**

The UM, JULES, and NEMO codes were build using the *fcm_make* extract and build system provided within the Flexible
Configuration Management (FCM) tools. UM, JULES, and WAVEWATCH III codes, and Rose suites, were also
configuration managed using this system. Further information is provided at http://metomi.github.io/fcm/doc/user_guide/

(last access: 27 October 2021). The WAVEWATCH III code was compiled using a simple bash script part of the controlling Rose suite.

**Data availability**

The nature of the 4-D data generated in running the various RCS-IND1 experiments at high resolution requires a large tape storage facility. These data are of the order of tens of terabytes in total (see Table 9). However, these data can be made available after contacting the authors. Each simulation namelist and input data are also archived under configuration management and can be made available to researchers to promote collaboration upon contacting the authors. Processed data

used in the production of figures in this paper are available via https://doi.org/10.5281/zenodo.5831575.

**Author contribution**

JMC is lead developer of the RCS-IND1 technical infrastructure, ran most of the simulations discussed in this paper, and wrote the system document aspects of this manuscript. HWL ran simulations with frictional heating, prepared figures, and provided discussion of the results in this manuscript. AMishra, AMitra and AG have implemented RCS-IND1 on MoES

HPC and have undertaken additional analysis of results. JP and AB developed the NEMO ocean configuration used in RCS-IND1 and AS developed the WAVEWATCH III wave configuration. All authors have contributed to discussion of the case study results in this paper through the WCSSP India project, including additional analyses across atmosphere, land, ocean and wave components. JMC and HWL led on the manuscript preparation with contributions from all co-authors.

**Competing interests**

The authors declare that they have no conflict of interest.

**Acknowledgements**

This work was conducted through the Weather and Climate Science for Service Partnership (WCSSP) India, a collaborative initiative between the Met Office, supported by the UK Government's Newton Fund, and the Indian Ministry of Earth Sciences (MoES). We are grateful for many fruitful discussions and interactions with many

researchers working across the partnership during the development and initial application of the RCS-IND1 configuration. We acknowledge the Indian Space Research Organisation (ISRO) for use of the INSAT satellite data in Figure 5 and 6.





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





| Configuration | RUN_MODE | RCS-IND1 suffix | Description |
|---|---|---|---|
| Fully Coupled | atm-ocn-wav | **aow** | Fully coupled atmosphere/land-ocean-wave simulation |
| Partially Coupled | atm-ocn | **ao** | Partially coupled atmosphere/land-ocean simulation, no wave interactions included in ocean or atmosphere |
| | atm-kpp | **ak** | Partially coupled atmosphere/land-KPP ocean simulation, no wave interactions included |
| | atm-wav | aw-o aw-c aw-oc aw-h aw-hc | Partially coupled atmosphere/land-wave simulation, no current effects in wave model, no SST or surface currents updated in atmosphere model. Different ocean forcing to atmosphere and wave models available (see Table 3 and 5). |
| | ocn-wav | ow-g ow-h | Partially coupled ocean-wave simulation. Different meteorological forcing data selected as 'g' or 'h' options (see Table 3). |

**Table 1**. **Summary of the RCS-IND1 fully and partially coupled configurations available within the RCS, and naming conventions used in this paper. Options are controlled in the RCS using the RUN_MODE environment variable.**
**Results for configuration names in bold are demonstrated in this paper. See also Figure 2.**





| Configuration | RUN_MODE | RCS-IND1 suffix | Description |
|---|---|---|---|
| Atmosphere/land only | atm | a **a-o** a-h | Regional atmosphere/land simulation. Different options for initialising and updating SST are available (see Table 3). |
| Ocean only | ocn | o-e o-g o-h | Regional ocean-only simulation. Different options for meteorological forcing are available (see Table 4). |
| Wave only | wav | w-g w-gc w-h w-hc | Regional wave-only simulation. Different options for meteorological forcing (see Table 4) and ocean current forcing (see Table 5) are available. |

**Table 2**. **Summary of the RCS-IND1 uncoupled configurations available in the RCS, and naming conventions used in this paper. Results from configuration names in bold are demonstrated in this paper. Options are controlled in the RCS using the RUN_MODE environment variable. See also Figure 2.**






| Order | Interface | Exchanged variable | Symbol | Units |
|---|---|---|---|---|
| 1 | W-A | Wave-dependent Charnock parameter | $\alpha$ | - |
| 2 | O-A | Sea surface temperature [+] | SST | K |
| 2 | O-A | Zonal surface current | $u_{curr}$ | m s$^{-1}$ |
| 2 | O-A | Meridional surface current | $v_{curr}$ | m s$^{-1}$ |
| 3 | A-O | Non-solar net surface heat flux [+] | $Q_{ns}$ | W m$^{-2}$ |
| 3 | A-O | Solar surface heat flux (all wavelengths) [+] | $Q_{sr}$ | W m$^{-2}$ |
| 3 | A-O | Rainfall rate [+] | $R$ | kg m$^{-2}$ s$^{-1}$ |
| 3 | A-O | Snowfall rate [+] | $S$ | kg m$^{-2}$ s$^{-1}$ |
| 3 | A-O | Evaporation of fresh water from the ocean [+] | $E$ | kg m$^{-2}$ s$^{-1}$ |
| 3 | A-O | Mean sea level pressure | $Pmsl$ | Pa |
| 3 | A-O | Zonal wind stress on ocean surface[*], [+] | $\tau_x$ | N m$^{-2}$ |
| 3 | A-O | Meridional wind stress on ocean surface[*], [+] | $\tau_y$ | N m$^{-2}$ |
| 4 | O-W | Zonal surface current | $u_{curr}$ | m s$^{-1}$ |
| 4 | O-W | Meridional surface current | $v_{curr}$ | m s$^{-1}$ |
| 5 | W-O | Significant wave height | $H_s$ | m |
| 5 | W-O | Zonal Stoke drift surface velocity | $u_s$ | m s$^{-1}$ |
| 5 | W-O | Meridional Stoke drift surface velocity | $v_s$ | m s$^{-1}$ |
| 5 | W-O | Mean wave period | $T_{01}$ | s |
| 5 | W-O | Zonal surface atmospheric stress transmitted to the ocean | $\tau_{wx}$ | N m$^{-2}$ |
| 5 | W-O | Meridional surface atmospheric stress transmitted to the ocean | $\tau_{wy}$ | N m$^{-2}$ |
| 6 | A-W | Zonal wind speed at 10 metres above surface | $U_{10}$ | m s$^{-1}$ |
| 6 | A-W | Meridional wind speed at 10 metres above surface | $V_{10}$ | m s$^{-1}$ |

**Table 3. Summary of the coupling exchanges between atmosphere/land (A), ocean (O), and wave (W) components within the RCS-IND1 regional coupled configuration. The fields marked with an asterisk are only exchanged in atmosphere/land-ocean coupled configurations. Only the fields marked with a cross are exchanged in atmosphere/land-KPP ocean coupled configurations.**





| SST_INIT | Description | Possible configurations |
|---|---|---|
| None | Initial condition SST in atmosphere boundary condition obtained from downscaling global-scale NWP (which uses OSTIA), and remains constant throughout simulation | a<br>aw-c |
| ostia | Initial condition SST in atmosphere boundary condition obtained from global-scale OSTIA on native ocean analysis grid and updated daily | **a-o**<br>aw-o<br>aw-oc |
| high_SST | SST in atmosphere model taken from km-scale resolution regional ocean model (e.g. IND1o) and updated hourly every day | a-h<br>aw-h<br>aw-hc |

**Table 4**. **Summary of the different SST initialisation and updating options available within the RCS, applicable for model configurations in which there is not dynamic coupling to an ocean model. Options are controlled using the SST_INIT environment variable.**

| UM_FORCING | Description | Possible configurations |
|---|---|---|
| file-core | Meteorological forcing from ERA5 analysis and applied in NEMO ocean model using bulk forcing algorithm. Fields are updated hourly. | o-e |
| flx-global | Meteorological forcing from Met Office operational global NWP data and applied in NEMO ocean model as direct flux forcing. Wave model forced with 10m winds from same NWP system. Radiation terms are updated every 3 hours and winds are updated hourly. | ow-g<br>o-g<br>w-g<br>w-gc |
| flx-high | Meteorological forcing from km-scale resolution regional atmosphere model (e.g. IND1a) and applied in NEMO as direct flux forcing. Wave model forced with 10m winds from regional atmosphere model. All variables are updated hourly. | ow-h<br>o-h<br>w-h<br>w-hc |

**Table 5**. **Summary of the different meteorological forcing options available within the RCS modelling framework, applicable for model configurations in which there is not dynamic coupling to a regional atmosphere model. Options are controlled using the UM_FORCING environment variable.**



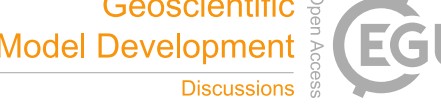

| WV_OCN_FORCING | Description | Possible configurations |
|---|---|---|
| false | No wave-ocean interactions are considered in a wave simulation | aw |
| | | aw-o |
| | | aw-h |
| | | w-g |
| | | w-h |
| true | Ocean currents from a regional ocean model (e.g. IND1o) are applied as forcing in a wave model, read in from external files, with fields updated hourly. | aw-c |
| | | aw-oc |
| | | aw-hc |
| | | w-gc |
| | | w-hc |

**Table 6. Summary of the uncoupled ocean forcing options available when running wave model configurations within the RCS modelling framework, applicable for model configurations in which there is not dynamic coupling to a regional ocean model. Options are controlled using the WV_OCN_FORCING environment variable.**

| Case | Initialisation | Duration | a-o | a-o | ak | ao | aow |
|---|---|---|---|---|---|---|---|
| Titli | 20181008 T00Z | 7 days | ATMfix | ATM | KPP | AO | AOW |
| Fani | 20190428 T00Z | 7 days | ATMfix | ATM | KPP | AO | AOW |

**Table 7. Summary of naming conventions used in describing RCS-IND1 simulation experiments for Titli and Fani cases. Simulations for which frictional heating is activated in the Unified Model configuration are signified by "_FH" after the relevant run identifier. ATMfix uses fixed SST for the duration of a simulation, while ATM has daily updating SST.**





| Case: | Titli | | | Fani | | |
|---|---|---|---|---|---|---|
| Summary metric: | Mean track error (km) | Minimum MSLP (hPa) [diff] | Maximum track wind speed (knot) [diff] | Mean track error (km) | Minimum MSLP (hPa) | Maximum track wind speed (knot) |
| **Observation** | - | **965** | **90** | - | **917** | **135** |
| ATMfix(_FH) | **52**** (58)_ | 933 (925) [-8] | **90** (98) [+8] | 103 (136) | 923 (**921**) [-2] | 100 (**105**) [+5] |
| ATM(_FH) | 74 (79) | 949 (948) [-1] | 86 (87) [+1] | 100 (132) | 926 (907) [-19] | 100 (104) [+4] |
| KPP(_FH) | 155 (131) | 953 (938) [-15] | 78 (88) [+10] | **85** (90) | 954 (936) [-18] | 86 (92) [+6] |
| AO(_FH) | 99 (83) | 969 (956) [-13] | 71 (82) [+11] | 114 (98) | 956 (951) [-5] | 90 (93) [+2] |
| AOW(_FH) | 88 (81) | **966** (951) [-15] | 65 (79) [+14] | 109 (104) | 959 (948) [-11] | 83 (98) [+15] |
| Sim. range | 103 (73) | 36 (31) | 25 (19) | 29 (46) | 36 (30) | 17 (13) |

**Table 8: Summary of observed and simulated cyclone statistics for Titli and Fani cases. Figures in round bracket italics indicate results from simulations with frictional heating enabled, with differences listed in square brackets. Underlined values indicate which of the simulations with or without frictional heating gives closest metric to observed for a given run type. Bold values indicate which simulation has best metric across all experiments conducted. The final row summarises difference between highest and smallest simulated values. [** Note ATMfix cyclone tracking**
**identified the cyclone later than for other experiments].**

| Configuration | a | o | w | ak | aw | ao | aow |
|---|---|---|---|---|---|---|---|
| Nodes used | 48 | 15 | 10 | 49 | 58 | 63 | 73 |
| Runtime/day[1] | 17 min | 20 min | 5 min | 18 min | 18 min | 20 min | 22 min |
| Runtime/day[2] | 16 min | 21 min | | 16 min | | 24 min | |
| Output/day | 15 Gb | 24 Gb | 3 Gb | 40 Gb | 30 Gb | 89 Gb | 109 Gb |

**Table 9. Summary of the typical computational resources required to run RCS-IND1 experiments, runtimes and output data volumes for completing a day simulation. Run durations quoted in row [1] were completed using the Met**
**Office Cray XC40 and those completed in row labelled [2] were completed using the NCMRWF High Performance Computing Server Mihir Cray XC40.**

**Figure 1: Illustration of RCS-IND1 domain coupled system domain extent. Shaded contours represent the**
**atmosphere model orography over land and ocean model bathymetry respectively. The region highlighted by the red box shows the region of focus for results presented in this paper. Marked locations indicate in-situ observation points referred to in the results section: from north to south, Red = Gopalpur [84.9E, 19.3N]; Magenta = 23091 [89.2E, 17.8N]; Yellow = 23093 [88.0E, 16.3N]; Blue = 23459 [87.0E, 14.0N].**



**Figure 2: Schematic summary of RCS-IND1 modelling framework configuration, experiment options and naming conventions. Configurations highlight with experiment identifiers in red are presented in this paper.**



**Figure 3: (a) Persisted SST field in ATMfix(_FH) control simulation, based on OSTIA data available for forecasting**
**at 00 UTC on 8 October 2018 for simulations of cyclone Titli, and (b)(c) initial condition difference in SST for (b)**
**KPP and (c) AO (and AOW) simulations. Note that ATMfix and ATM use same SST on first day of simulations, and**
**that initial conditions are common to each experiment with/without frictional heating. (d) Timeseries of regional**
**mean SST for each experiment during 7-day Titli case study, for the sub-domain shown in (a). (e)-(h) Difference in**
**SST for ATM_FH, KPP_FH, AO_FH and AOW_FH simulations respectively at run final time compared to run start**
**time, illustrating extent and magnitude of cooling during each simulation. (i)(j)(k) Comparison of timeseries of SST**
**from each experiment with in-situ ocean buoy observations at 23459, 23093 and 23091 respectively. Model data are**
**taken as means in 5x5 grid neighbourhood surrounding each buoy location (shown in Figure 1 and panel (a)).**
**Simulations without frictional heating are shown as solid lines, with frictional heating with dashed lines.**







**Figure 4: (a) Persisted SST field in ATMfix(_FH) control simulation, based on OSTIA data available for forecasting at 00 UTC on 28 April 2019 for simulations of cyclone Fani, and (b)(c) initial condition difference in SST for (b) KPP and (c) AO (and AOW) simulations. Note that ATMfix and ATM use same SST on first day of simulations, and that initial conditions are common to each experiment with/without frictional heating. (d) Timeseries of regional mean SST for each experiment during 7-day Fani case study, for the sub-domain shown in (a). (e)-(h) Difference in SST for ATM_FH, KPP_FH, AO_FH and AOW_FH simulations respectively at run final time compared to run start time, illustrating extent and magnitude of cyclone-induced cooling captured in each simulation. (i)(j)(k) Comparison of timeseries of SST from each experiment with in-situ ocean buoy observations at 23459, 23093 and 23091 respectively (locations shown in Figure 1 and panel (a)). Model data are taken as means in 5x5 grid neighbourhood surrounding each buoy location. Simulations without frictional heating are shown as solid lines, with frictional heating with dashed lines.**





**Figure 5: (a) Illustration of Cyclone Titli satellite observed brightness temperature from Meteosat at 00 UTC on 10 October 2018. In (b)-(f), results from RCS-IND1 experiments showing maximum wind speed during the 7-day simulation (coloured shading). Also plotted is the diagnosed simulated cyclone track at 3-hourly intervals (coloured line and squares) compared with observed track at 6-hourly intervals (black line and circles). Line contours show instantaneous simulated outgoing longwave radiation at top of atmosphere (dashed line indicating 150 Wm$^{-2}$, solid line 100 Wm$^{-2}$ contour) at coincident hour to the satellite observation in (a). See Figure S1 for corresponding figure for simulations without frictional heating enabled.**

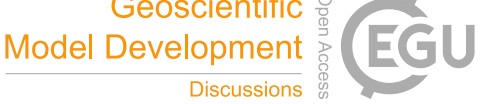


**Figure 6: (a) Illustration of Cyclone Fani satellite observed brightness temperature from INSAT at 06 UTC on 2 May 2019. In (b)-(f), results from RCS-IND1 experiments showing maximum wind speed during the 7-day simulation (coloured shading). Also plotted is the diagnosed simulated cyclone track at 6-hourly intervals (coloured line and squares) compared with observed track at 6-hourly intervals (black line and circles). Line contours show**

**instantaneous simulated outgoing longwave radiation at top of atmosphere (dashed line indicating 150 Wm$^{-2}$, solid line 100 Wm$^{-2}$ contour) at coincident hour to the satellite observation in (a). See Figure S2 for corresponding figure for simulations without frictional heating enabled.**



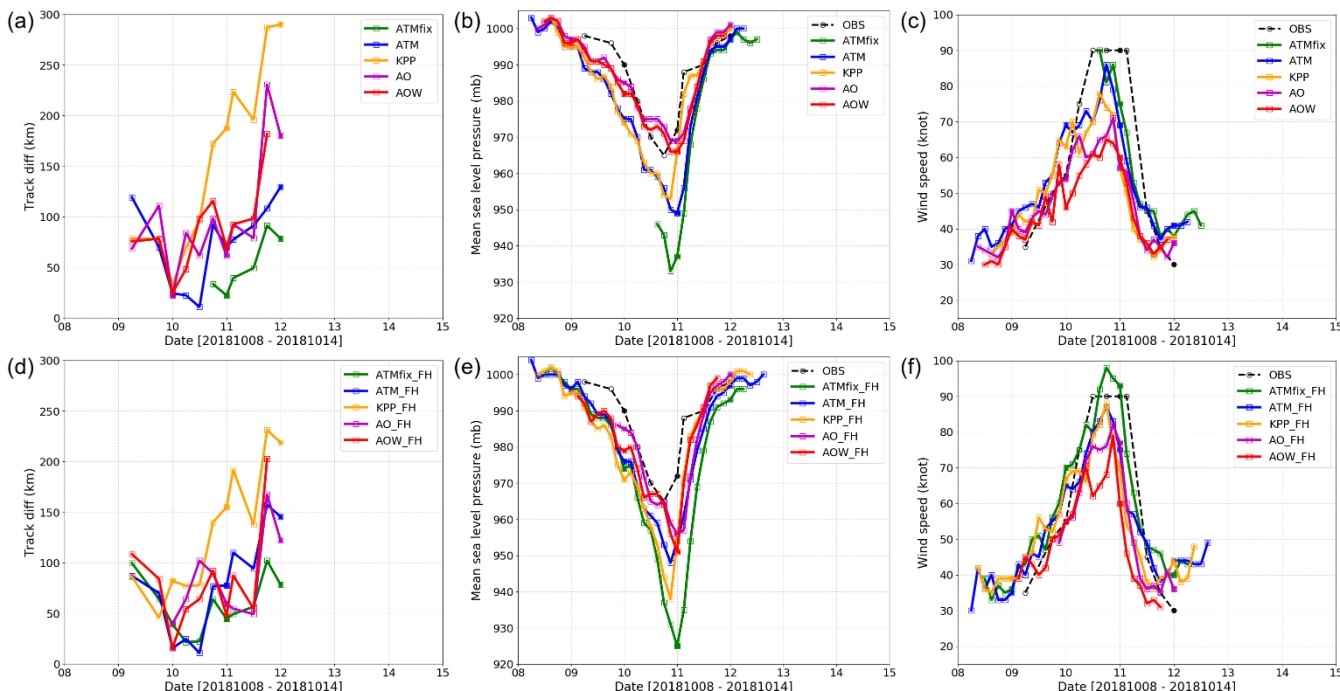

**Figure 7: Comparison of observed and simulated cyclone track statistics for all simulations of cyclone Titli. Panels (a)-(c) show results without and (d)-(f) with frictional heating represented. Panels (a) and (d) show difference between coincident simulated and observed track position, (b) and (e) show mean sea level pressure at diagnosed track centre and (c) and (f) show along-track maximum wind speed. Observed track data are taken from the SXXT50 bulletin based on satellite data interpretation.**



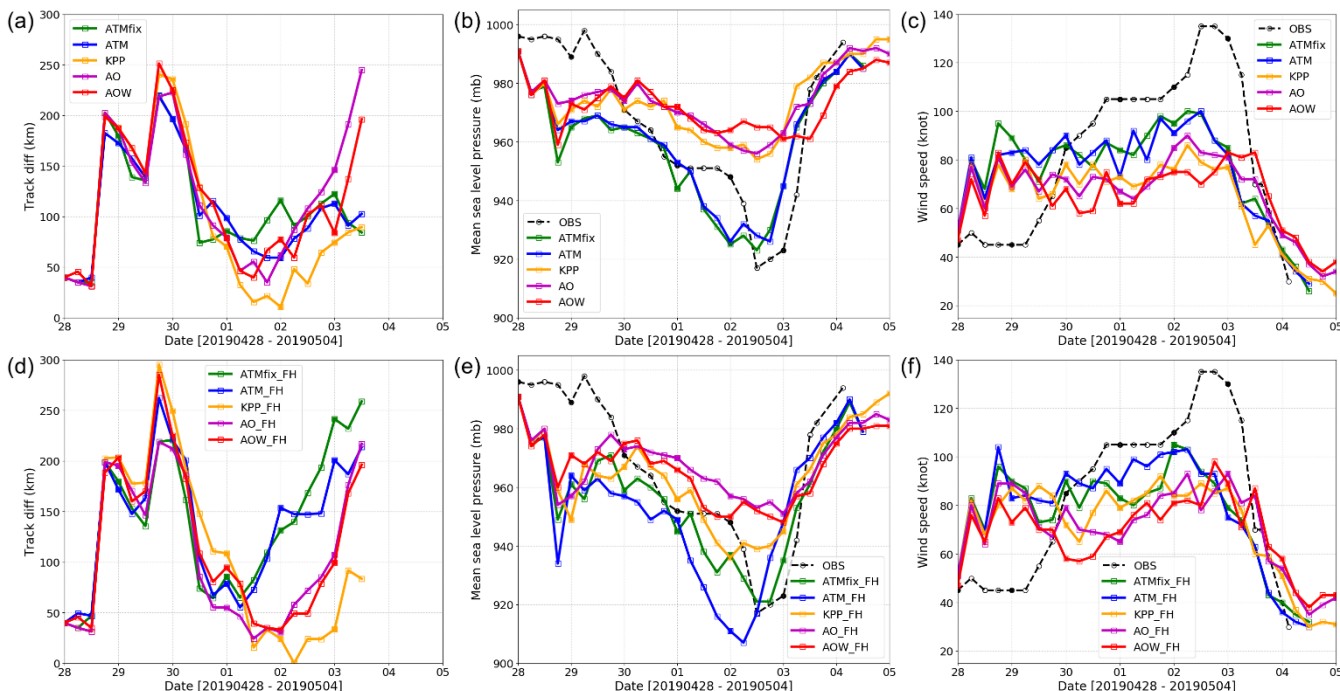

**Figure 8: Comparison of observed and simulated cyclone track statistics for all simulations of cyclone Fani. Panels (a)-(c) show results without and (d)-(f) with frictional heating represented. Panels (a) and (d) show difference between coincident simulated and observed track position, (b) and (e) show mean sea level pressure at diagnosed track centre and (c) and (f) show along-track maximum wind speed. Observed track data are taken from the SXXT50 bulletin based on satellite data interpretation.**



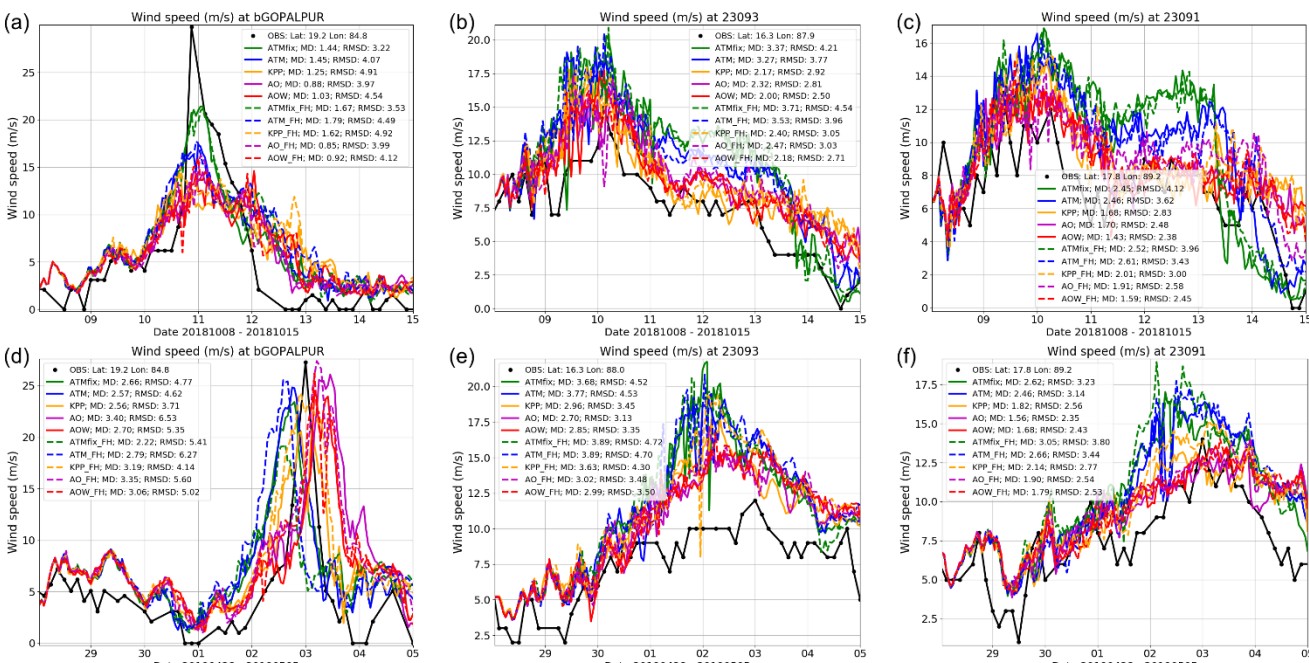


**Figure 9: Comparison of observed (black lines) and simulated wind speed at (a)(d) Gopalpur coastal location, (b)(e) 23093 ocean buoy and (c)(f) 23091 ocean buoy during (a)-(c) Titli and (d)-(f) Fani case study periods. See Figure 1 for summary of observation locations. Note different y-axis scales are used in each panel. Simulation results without (solid lines) and with frictional heating (dashed lines) are provided, based on mean of 5 x 5 neighbourhood of grid**
**points nearest the observation point.**



**Figure 10: 7-day precipitation accumulation during Titli case study over Bay of Bengal sub-domain region (a) observed by GPM (NASA Global Precipitation Measurement), and simulated by (b) ATMfix_FH, (c) ATM_FH, (d) KPP_FH, (e) AO_FH and (f) AOW_FH. All simulated data are interpolated to the GPM resolution prior to computing accumulations. See Figure S3 for corresponding figure for simulations without frictional heating enabled.**

**Figure 11: 7-day precipitation accumulation during Fani case study over Bay of Bengal sub-domain region (a) observed by GPM (NASA Global Precipitation Measurement), and simulated by (b) ATMfix_FH, (c) ATM_FH, (d) KPP_FH, (e) AO_FH and (f) AOW_FH. All simulated data are interpolated to the GPM resolution prior to computing accumulations. See Figure S4 for corresponding figure for simulations without frictional heating enabled.**



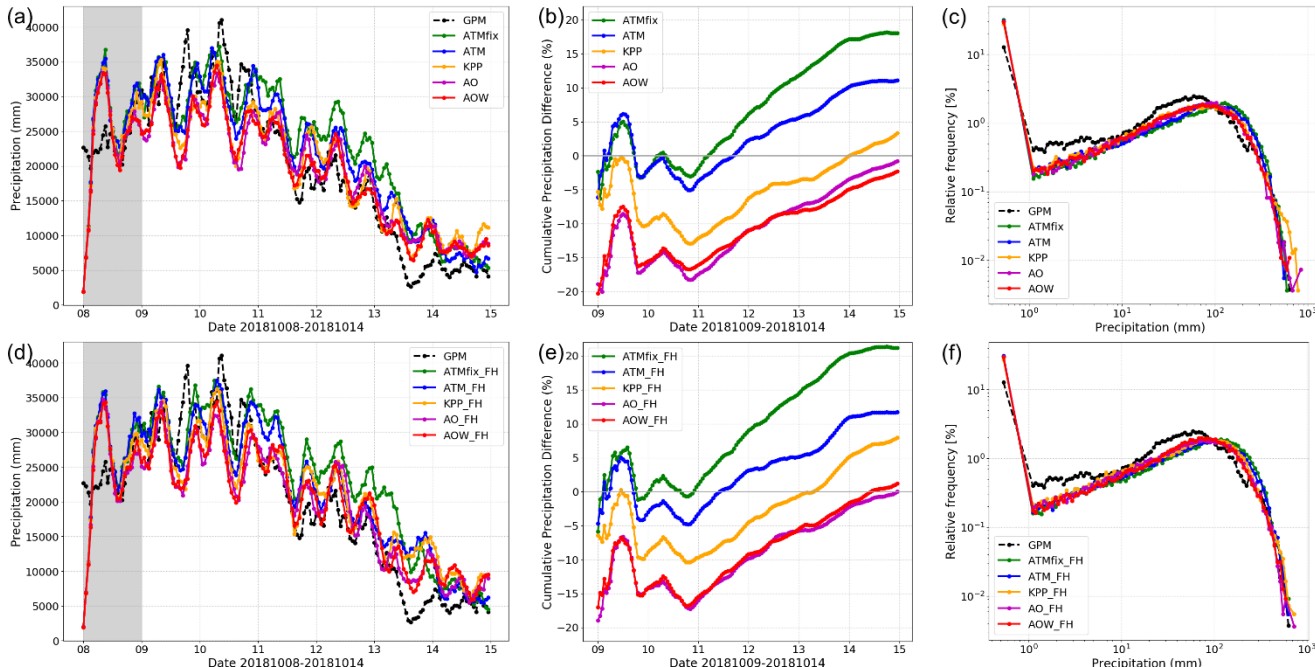

**Figure 12: Comparison of Bay of Bengal sub-domain precipitation characteristics for Titli case study for simulations (a)-(c) without and (d)-(f) with frictional heating enabled. (a)(d) Timeseries of sub-domain accumulated precipitation for region shown in Fig. 10. (b)(e) Cumulative % difference between simulated precipitation and GPM observations, computed after day 1 of simulation to avoid spin up effects. (c)(f) Frequency distribution of 7-day accumulated precipitation relative to GPM.**





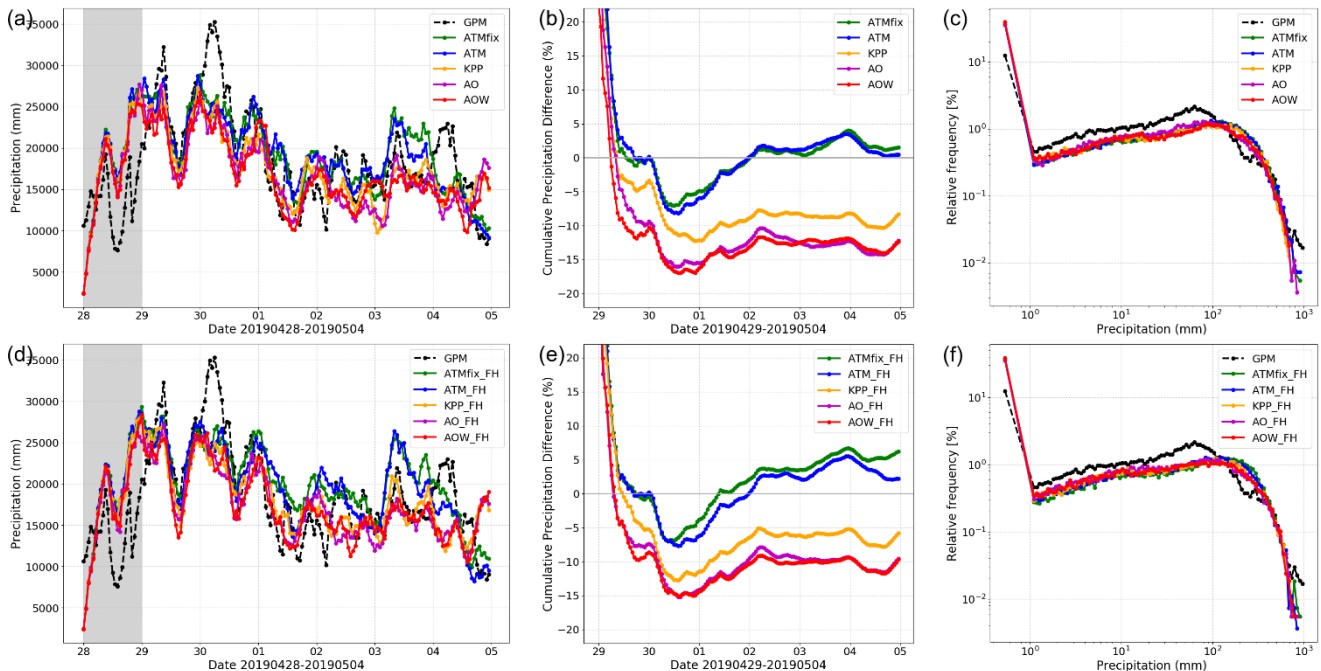

**Figure 13: Comparison of Bay of Bengal sub-domain precipitation characteristics for Fani case study for simulations (a)-(c) without and (d)-(f) with frictional heating enabled. (a)(d) Timeseries of sub-domain accumulated precipitation for region shown in Fig. 10. (b)(e) Cumulative % difference between simulated precipitation and GPM observations, computed after day 1 of simulation to avoid spin up effects. (c)(f) Frequency distribution of 7-day accumulated precipitation relative to GPM.**