# Peer review of "The Regional Coupled Suite (RCS-IND1): application of a flexible regional coupled modelling framework to the Indian region at kmscale"

_Geoscientific Model Development, 2022_

## Referee Comment (RC1)

**Title:** "The Regional Coupled Suite (RCS-IND1): application of a flexible regional coupled modelling framework to the Indian region at kmscale"

**Summary**

Using a kmscale high-resolution regional climate model, the authors examined two tropical cyclone cases (Titli in Oct 2018 and Fani in April 2019) in the Bay of Bengal. Especially, five different simulations have been performed based on combinations of different components of the regional climate model. Major finding is that air-sea coupling can improve the simulated Typhoon track and intensity. It is also shown that the vertical turbulent mixing (K-profile parameterization, KPP) and frictional heating are important for the variation and intensity of air-sea variables during Typhoon period.

**Recommendation:** Accept after minor revisions.

The paper is generally quite well written, despite the minor points below. The authors have presented a clear description of model configuration and a convincing set of experiments in simulating the sea surface temperature, Typhoon characteristics and precipitation by comparing them with observations. In terms of kmscale high-resolution, the regional coupled model resolves the limitation of local parameterization (e.g., turbulent mixing, convective process) and provides a more realistic presentation of physical process and also saves computation resources (compared to global coupled model). The study would be of relevance and interest to the regional coupled model community as well as to the Typhoon-related studies in regional areas.

Here are some specific points below:

**Major point:**

The choice of boundary condition is a challenging issue for regional modelling. As shown in Figure 1, **the southern boundary of OCN** may not include the whole path of the Monsoon Currents which exchange water and mass between Bay of Bengal and Arabian Sea (Schott et al., 2001), and the equatorial undercurrent current, which may transport water and high-salinity into the Bay of Bengal (Matthews et al., 2019). Hence, there is a possibility that influence of those currents to the Bay of Bengal is underestimated. Please explain the above-mentioned issue. Also, please add longitude and latitude axises to Figure 1.

References:

Schott, F. A., & McCreary Jr, J. P. (2001). The monsoon circulation of the Indian Ocean. Progress in Oceanography, 51(1), 1-123.

Sanchez-Franks, A., Webber, B. G. M., King, B. A., Vinayachandran, P. N., Matthews, A. J., Sheehan, P. M. F., et al. (2019). The railroad switch effect of seasonally reversing currents on the Bay of Bengal high-salinity core. Geophysical Research Letters, 46, 6005–6014. https://doi.org/10.1029/2019GL082208

**Minor points:**

1. Line 370, please explain how this "frictional heating" term is calculated. Perhaps a formula would be better for the readers to understand the related physical process.

2. Storm surge disaster caused by Typhoon is another important issue. It is found that the sea level variation is mainly related to tide, wind and topography-induced friction and so on (Xuan et al., 2021). Since this study included those factors, it would be interesting to find out the performance of sea level variation (in terms of periodicity and correlation) under kmscale coupled framework, especially for the AO and AOW cases. Although the authors left this topic for future research in the discussion, it is related to your current study and can be used to evaluate your main results (wind, tide). So, I'd be glad to see some analysis at least in the supplementary information, if feasible.

Reference:

Xuan, J., Ding, R., and Zhou, F. (2021). Storm surge risk under various strengths and translation speeds of landfalling tropical cyclones. Environmental Research Letters. doi: 10.1088/1748-9326/ac3b78.

---

## Author Comment (AC1)

**Author comments on:**

**The Regional Coupled Suite (RCS-IND1): application of a flexible regional coupled modelling framework to the Indian region at km-scale**

Juan M. Castillo[1], Huw W. Lewis[1], Akhilesh Mishra[2], Ashis Mitra[2], Jeff Polton[3], Ashley Brereton[3], Andrew Saulter[1], Alex Arnold[1], Segolene Berthou[1], Douglas Clark[4], Julia Crook[5], Ananda Das[6], John Edwards[1], Xiangbo Feng[7], Ankur Gupta[2], Sudheer Joseph[8], Nicholas Klingaman[7], Imranali Momin[2], Christine Pequignet[1], Claudio Sanchez[1], Jennifer Saxby[5], Maria Valdivieso da Costa[7]

[1]Met Office, Exeter, EX1 3PB, UK
[2]National Centre for Medium Range Weather Forecasting (NCMRWF), India
[3]National Oceanography Centre, Liverpool, UK
[4]UK Centre of Ecology & Hydrology (UKCEH), UK
[5]University of Leeds, UK
[6]India Meteorological Department (IMD), India
[7]University of Reading, UK
[8]INCOIS, India

*Correspondence to*: Juan M. Castillo (juan.m.castillo@metoffice.gov.uk)

**Summary**

We are grateful to the 3 anonymous reviewers who gave their time to consider the submitted manuscript and for their insightful comments and positive responses. We kindly acknowledge that all reviewers commended the paper content and relevance and agreed that the manuscript should be accepted for publication following minor revisions. The specific comments received will be addressed in a revised manuscript with tracked changes that will be submitted on finalising this author response.

A detailed summary of responses to all review comments is provided below, including references to amendments that are included in the revised manuscript. Original review comments are shown in red for clarity, with author comments in black.

**RC1**

Major point: The choice of boundary condition is a challenging issue for regional modelling. As shown in Figure 1, the southern boundary of OCN may not include the whole path of the Monsoon Currents which exchange water and mass between Bay of Bengal and Arabian Sea (Schott et al., 2001), and the equatorial undercurrent current, which may transport water and high salinity into the Bay of Bengal (Matthews et al., 2019). Hence, there is a possibility that influence of those currents to the Bay of Bengal is underestimated. Please explain the above-mentioned issue.

We agree with the reviewer's perspective here, and indeed choice of the domain southern boundary, from ocean perspective was one of the key considerations ahead of establishing the RCS-IND1 domain. This was not very explicitly captured in the manuscript (only passing reference in line 135 to

capturing equatorial ocean currents). A short additional paragraph will be provided in updated manuscript (from line 136) to expand on these considerations. In practice, the option would be to aim for a southern boundary sufficiently far north that the transports due to equatorial currents can be fully captured through the lateral boundaries, or to extend the RCS-IND1 domain quite far south (e.g. to at least 3S) in order to fully include monsoon currents and equatorial currents within the domain (albeit with increased dependence on east-west lateral boundaries). To limit the domain extent, the former option was adopted, following the lead of partners at INCOIS and development of their ROMS-based regional systems (Francis et al., 2020; Remya et al., 2020).

Please add longitude and latitude axes to Figure 1.
Figure 1 will be revised in updated manuscript with lat/lon labels as suggested, and updated choice of colour scales (see RC3). See also revised Figure 1 further below.

1. Line 370, please explain how this "frictional heating" term is calculated. Perhaps a formula would be better for the readers to understand the related physical process.
The description of the UM boundary layer formulation for frictional heating (i.e., heating increment from turbulence dissipation) will be expanded in revised manuscript, including an additional equation (2) as suggested. The approach follows Zhang and Altshuler (1999) [Mon. Wea. Review; their eq. 6] in modifying the thermal energy equation. See also similar request from RC2.

2. Storm surge disaster caused by Typhoon is another important issue. It is found that the sea level variation is mainly related to tide, wind and topography induced friction and so on (Xuan et al., 2021). Since this study included those factors, it would be interesting to find out the performance of sea level variation (in terms of periodicity and correlation) under km-scale coupled framework, especially for the AO and AOW cases. Although the authors left this topic for future research in the discussion, it is related to your current study and can be used to evaluate your main results (wind, tide). So, I'd be glad to see some analysis at least in the supplementary information, if feasible.
We agree with the reviewer that storm surge prediction is a key impact of tropical cyclones, and also an important driver of developing regional coupled capabilities to support enhanced multi-hazard predictions (e.g. line 82).
Some analysis of predicted storm surge using RCS-IND1 has been conducted (by co-author Feng) for Fani and Titli cases, but for runs without frictional heating only at this point. While the authors are encouraged by the reviewer's suggestion to include this analysis, at least in supplementary information, it is our view that an adequate discussion of ocean hazard metrics would considerably increase the manuscript length (and noting already 13 proposed Figures and 4 additional supplementary figures). For example, it becomes necessary to introduce discussion of relevant ocean-only control simulations (in same way that a number of atmosphere-only control simulations are introduced in the material presented in the submitted manuscript). For reference, the figures below (courtesy co-author Feng), illustrate the maximum storm surges predicted for TC Fani by AO and AOW simulations, along with ocean-only simulations (IND1_o-e forced by global ERA5 meteorology, IND1_o-g forced by global UM meteorology and IND1_o-h forced by an atmosphere-only 4.4 km regional simulation). The comparison with surge observations at Vishakhapatnam (starred location in surge maps) indicates that systems forced with 4.4 km resolution meteorology (including AO and AOW) capture the order of magnitude of observed surge (~40 cm), but the timing is too early, thought to be related to errors in the predicted tracks. Sensitivity to coupling is apparent.

[Figure]

**Reference figure:** (left) illustration of maximum simulated storm surge for TC Fani case (all runs without frictional heating), and (right) comparison of computed storm surge with observations (black) from Vishakhaptnam tide gauge (location marked in left panel with star).

It remains the author's consideration that a fuller analysis of storm surge results from across a number of storm cases, with appropriate discussion and justification of experimental design for ocean hazards is outside the useful size of the current manuscript, and worthy of fuller discussion in a separate paper.

**RC2**

Line 63: "focussed" is used in the rest of manuscript
Corrected in revised manuscript.

Lines 189-191: I am trying to understand this advantage. Please give an example and provide more details, because the air-sea momentum transfer is a very important factor in cyclones.
In the approach previously documented by Lewis et al. (2019; see their Equation 3, and their Table 3), the coupling variable between the wave and ocean model used was 'tauoc' – i.e., the fraction of atmospheric stress transferred to the ocean. In practice, this could lead to some inconsistencies with the surface stress being computed in NEMO, WAVEWATCH and UM/JULES (each using their own surface parameterisation), with tauoc being computed in WAVEWATCH based on its wind-to-stress calculation, and accounting for wave processes, but then applied in NEMO to compute the modified surface stress based on the atmospheric stress components coupled directly from UM/JULES to NEMO. Within RCS-IND1 (see manuscript Table 3), the atmosphere model computed stress is only passed to NEMO in AO mode, whereas in AOW mode, the wave-modified atmosphere stress components are transferred directly. Line 189-191 was aiming to highlight the more self-consistent treatment enabled by explicit transfer of stress terms. This argument will be clarified in the revised manuscript. Note that further simplifications to the surface momentum transfer in 3-way coupled system are being explored.

Line 210: Does the time step refer to the atmospheric model or to the land surface model?
For a resolution of 4.4 km, an atmospheric time step of 120 s sounds large. Please clarify
it.

The manuscript will be revised to confirm this is the UM (atmosphere) and JULES (land) model timestep. This matches the timestep of the operational NCUM-R (e.g. https://www.researchgate.net/publication/350061967_NCUM_Regional_Model_Version_4_NCUM-R_V4). Note that the initial NCUM-R implementations used a shorter timestep of 60 s (e.g. Mamgain et al., 2017). Some assessment of the influence of model timestep on TC simulation has been conducted using RCS-IND1 (for TC Amphan), but this is considered out of scope of this documentation paper.

Line 227: Improve -> Improvement of
Will be amended in updated manuscript.

Section 2.6: Please provide information about the bathymetry used in the wave model.
The wave component uses the same bathymetry definition as derived for the ocean model, and will be clarified in updated manuscript.

Line 315: How did you choose the upper limit of 0.32 for Charnock?
In UKC3, there was no upper cap to the computed Charnock value, and very large instantaneous values were sometimes possible. Note that the cap of 0.32 is an order of magnitude greater than typical climatological values (e.g. see Lewis et al., 2019; Figure 2(g) for UK-focussed regional climatology). This will be clarified in updated manuscript.

Lines 373-374: I suppose that with the term "frictional heating" you mean dissipative heating. It is usually considered as a term that added in sensible heat flux calculation in surface layer parameterizations of atmospheric models. How do you estimate it in your model? E.g., provide an equation.
The description of the UM formulation for frictional heating (i.e., heating increment from turbulence dissipation) will be expanded in revised manuscript, including an additional equation (2) as suggested. The approach follows Zhang and Altshuler (1999) [Mon. Wea. Review; their eq. 6] in modifying the thermal energy equation. See also similar request from RC1.

486-487: MSLP differences of 30-36 hPa seem very large. I think that the SST cooling presented in this study may hardly result in such large pressure differences? Do they agree with MSLP differences reported by other studies using coupled systems for tropical cyclones?
We agree that these differences do seem very large (i.e., RCS-IND1 simulations of TC are very sensitive to ocean state). This will be more clearly highlighted in the revised manuscript. It should however be noted that SST differences between ATMfix and AOW of up to 2 K develop, and that the ocean state in coupled simulations are initialised generally cooler than uncoupled simulations initialised from analysis (i.e. Figures 3 and 4). In this context, Rai et al. (2018), based on use of different resolution of (time-varying) SST products for atmosphere-only simulations of TC Phailin in Bay of Bengal found even a 0.2 – 0.4 K cooler SST around the TC centre could result in around 7 hPa less intense storm after 78 h forecast time. The reason for such large sensitivity would be interesting to explore in further work – for example to determine the extent to which ocean and/or atmosphere model resolution is particularly important (other studies are typically running with coarser regional or global model resolutions). If reviewers have particular studies in mind for expecting much smaller sensitivity, we would be happy to reflect this in a revised manuscript.

Line 498: Why does AOW result in slightly earlier intensification? The increased wave induced sea surface roughness in AOW is expected to delay the intensification due to the kinetic energy loss in the surface layer. Please explain your finding.

A physical explanation has not been offered alongside description of the AO vs AOW differences in the draft manuscript, considering it being unclear if this is a systematic feature, or an observation of the difference in this case. We note (and will reflect this in revised manuscript) that AO and AOW take different cyclone paths around the time of the different intensification, particularly in runs with frictional heating, with AO deviating westward relative to AOW. In practice, it is challenging to attribute a simple cause/effect directly for the differently evolving TCs with a number of contributing factors. Assessing the impact of wave coupling, relative to AO, for a number (e.g., 10+) of cases may be more instructive for determining general characteristics.

Line 510: is however -> are however

Modified in revised manuscript.

Line 535: "having increased MSLP". Maybe, do you mean "decreased"?

The intended context of this line was in discussing "increased *errors* of MSLP", but we agree this is not very clear as written. This sentence is revised in the updated manuscript, to read instead *"...with ATMfix_FH and ATM_FH over-deepening and thereby having larger errors of MSLP and track position relative to the equivalent ATMfix and ATM simulations without frictional heating."*

Line 554-557: How is this inconsistency in MSLP and maximum wind speed explained?

This section will be revised slightly in the updated manuscript to clarify the discussion.

This discussion relates to an apparent improvement in the wind-pressure relationship with coupling found over number of cases - for a given central MSLP, coupled simulation peak wind speeds are stronger and thereby in closer agreement to observed peak wind speed for that given MSLP. To some extent therefore, this represents greater consistency (of the coupled system) rather than an *inconsistency*. It is a bit unfortunate, but not clear to authors how to rectify, that the discussion of the impact of drag parameterisation in RAL1-T is not introduced until later in this section (from line 580 in original manuscript), as this helps to explain why we find a growing slow wind speed bias with deeper simulated storm systems (see also RC2 comment re. lines 672-674). The situation is apparently improved in the coupled system, but it is not currently clear, without further analysis across a number of cases, if this is more a symptom of simulating generally less deep storms (i.e. MSLP increased, for same simulated wind speed). This question would need to be examined in context of simulating deeper storms with the coupled system (i.e., if AOW were able to deepen to ~920 hPa, does this generate stronger peak winds in excess 100 kn?).

It should be emphasised that Fig. 7 and 8 also show, in general, consistent behaviour between MSLP and maximum wind speed between different simulations (i.e., deeper simulated storm, stronger peak winds).

Lines 555 & 557, Table 8 and Figures 7 & 8: m/s is preferable than knots to be consistent with Figures 5, 6 and 9.

This will be corrected in the revised manuscript – Figures 7(c),(f) and Figures 8(c),(f) updated to be in m/s (not replicated here for brevity), and values quoted in text and Table 8 all in m/s.

Line 575: You mention that the impact of wave coupling on wind speeds is relatively small. However, according to relative studies using coupled systems, wave coupling seems to have strong effects on momentum exchange and, subsequently, on wind speed because

it changes the roughness length and the drag coefficient. I appreciate your discussion in
L581-592 about sea surface drag and the decrease of drag coefficient in high intensities,
but please further explain the finding presented in L575. Also, write in the text a range of
wind speed differences between the simulations because the color palette does not help
the readers to quantify the differences.

A further line of text is inserted in the revised manuscript to highlight that the improvement of winds relative to Gopalpur observations with coupled frictional heating runs (generally with peak winds earlier and greater magnitude), is consistent with a faster storm translation and deeper systems developing when we include an additional source of near-surface heating. On the relative importance of wave coupling, we find much greater sensitivity on wave coupling (and drag parameterisation) for mid-latitude storm systems (e.g., Gentile et al. 2021).

Discussion of the quantitative wind speed differences between simulations are mostly focussed on discussion of track characteristics (Fig 7. and 8.; Table 8), as maximum wind speed shown in Fig 5 and 6 generally follow the diagnosed cyclone track. However, for reference, maximum model wind will be quoted on updated Fig 5 and 6 panels (not reproduced here for brevity), with values as follows consistent with those provided in Table 8:

TITLI (Fig. 5):
ATMfix_FH: 49.1; ATM_FH: 45.2; KPP_FH: 45.4; AO_FH:  42.2; AOW_FH: 40.5 m/s

FANI (Fig. 6):
ATMfix_FH: 56.5; ATM_FH: 55.7; KPP_FH: 49.9; AO_FH:  49.4; AOW_FH: 50.3m/s

Lines 595-628: It is a little unclear for me which simulation has the best overall
performance. Putting it another way, which coupled configuration would you choose to
better predict rainfall during TCs in the India region? An approach using contingency table
and respective statistics for discrete variables could support the evaluation.

The reviewer makes a valid request here – which configuration gives best overall performance? The Discussion of the revised manuscript will expand on why this has not been the evaluation focus for this paper (rather, introducing RCS-IND1 and demonstrating the flexible approach to running experiments). The key barrier at present is the different initialisation approaches required for the ocean state across coupled and uncoupled configurations, which makes attribution of 'best' a challenge given that ATM and KPP simulations are initialised from analysed SST, while AO and AOW simulations are initialised from free-running ocean-only multi-annual spin-up simulations. The evaluation proposed by the reviewer would be better suited to a future study, in which the coupled system initialisation can be more comparable to the analyses and looking across a broader number and range of cases to build up a more reliable set of statistics. Note this limitation was discussed from line 397 of the original manuscript.

Lines 672-674: Do you use a drag formulation including saturation in very high wind
speeds? Such formulation could impact not only momentum exchange but also heat
exchange through the change of Ck (bulk air-sea enthalpy transfer coefficient). Please
provide more information about these important processes in the surface layer.

The drag formulation used in the RCS-IND1 configurations described in this paper do no have saturation at very high wind speeds, and we agree that the details of the parameterisation of surface drag are key to the system performance and relative impact of coupling. The discussion on the details of the surface drag provided in the original manuscript (from line 580) will also be highlighted in the revised Discussion section of the updated manuscript. We also refer to linked work by Gentile et al

 as illustrative of ongoing evaluation of the different options for representing drag (e.g., capping at higher wind speeds, as used in subsequent versions of the RAL physics definitions).

Table 9: Please check output/day values, they seem inconsistent. For example, AOW resulted to 109 Gb/day, but summing a, o and w gives 42 Gb/day.

Table 9 will be updated in the revised manuscript in light of this comment, as we agree that the presentation of output/day totals is not clear as presented. The revised Table 9, copied below for reference, distinguishes between the 'Diagnostic' Output/day (i.e. data volume typically archived and required for simulation analysis and model dumps for restarting), and the 'Coupling' Output/day (i.e. difference between total output written to disk and Diagnostic output, indicative of the additional data written to disk for coupling). In the reviewed manuscript, the Total Gb/day were quoted (i.e. Diagnostic + Coupling), and thereby AOW has a much greater data usage reflecting 3-way coupling exchanges between components, in addition to the ~40Gb/day diagnostic output (which does just sum in the way the reviewer might expect). We trust the revised Table 9 provides a more useful synthesis by separating these two contributions to the output/day totals.

Figures 12 & 13: Although the spatially accumulated precipitation expressed in mm can be used for the comparison of simulations results, it is dependent on horizontal spacing used and, thus, it does not have robust physical meaning. For example, if you used 2 times higher resolution you would have 4 times higher spatially accumulated precipitation values, given the same area. So, it would be better to express the spatially accumulated precipitation as kg (or tons) per total area instead of mm. Another approach would be the estimation of areal precipitation which is the average precipitation depth over the area.

All precipitation diagnostics presented in the draft manuscript are derived from grid box average precipitation fluxes (mm/h from GPM, $kg/m2/s$ from model outputs). The accumulations presented therefore represent the cumulative hourly average precipitation depth over the area shown, noting all outputs are first interpolated to the same GPM output grid prior to comparison. We consider the presentation of precipitation results to be robust, and suitable for comparison between simulations, as reflected by the reviewer comments. This also follows the methodology and precipitation definitions used, for example, in generation of national-scale climatologies, also expressed in mm. The definition of accumulated precipitation will be clarified in the revised manuscript.

**RC3**

Line 46-56: I would move this part at the end of the introduction.

The authors appreciate the suggestion, and consider the best approach may be to split these lines between those that provide useful context to following Introduction discussion (i.e. lines 46-49), while the more system-specific part of this paragraph will be blended with lines 93 onwards. This also helps to avoid some repetition of themes between those paragraphs.

Line 85- Chlorophyll-*a*

This is amended in the revised manuscript.

Line 113 where, required,..

We disagree with this suggested grammatical edit, but the sentence has been split and simplified for brevity.

Line 123-125 I do not understand very well this statement. Do you mean that Jules behaves as a library of the UM? Please explain better

The UM atmosphere and JULES land surface model are compiled as a single executable, although the codes are separate. This supports the implicit coupling approach described by Best et al. (2004). This approach is common to many land surface models that can run independently or coupled directly to an atmosphere code. This sentence is clarified in the revised manuscript.

Section 2.1, section 2.3 and section 2.4 : I would merge these sections in just one. In this way a potential Reader would not need to jump from section 2.1 to section 2.3 to get information about the vertical resolution of the ocean model NEMO (as I did)

We agree that there is some 'jumping around' on key information between these sub-sections. However, the direct discussion of choice of model domain both helps to set the context, and is relevant to comments of RC1 on appropriate balance between requirements of atmosphere and ocean domains. The aspects of Section 2.1 that reference details of the model grid are moved to the relevant sub-sections however, to simplify this discussion in the revised manuscript.

We disagree with the suggestion to merge these sub-sections entirely however, as this would lead to an overly long and difficult to follow section of model documentation.

Fig.1 the two colorbars share some colors (for example the blue). This could lead to some confusion in reading the Figure 1. I would suggest to redraw the figure 1 with different colorbars.

Figure 1 will be revised in updated manuscript with updated choice of colour scales as suggested, along with lat/lon labels (see RC1). See also revised Figure 1 further below.

Line 276 What do you mean with "multi-annual"…please specify.

We agree this is ambiguous. As stated (more clearly) in remainder of the sentence, the regional ocean was initialised using a long simulation, initialised from rest and T&S interpolated from a global ocean product running from 1 January 2016. For the TC Fani case, this configuration had therefore been run for 3 years prior to the case study initial conditions. The phrase "multi-annual" is superfluous and has been removed from the revised manuscript.

Figure 3-4 Maybe using oC would make the maps and graphs more readable. I would also use different markers and colors for the location of the buoys. Did you test if the differences observed in the maps are statistically significant or not? This question holds also for other figures where you compare observations and simulated fields.

We appreciate the suggestion to work in °C rather than K for temperature but have consistently adopted SI units of K in both figures and text discussion, and thereby suggest to continue with this approach. Note that the central scale value of 303 K in Fig 3(a) and Fig 4(a) is approx.. 30°C – this will be highlighted in revised figure caption. We will also submit amended versions of Fig 3 and Fig 4 with the revised manuscript (not copied here for brevity) in which the colour marking for 23093 buoy location in central Bay of Bengal does not merge with the selected temperature colour scale – the proposed buoy marker colour changes are reflected in the revised Figure 1 copied below.

On comment re. statistical significance, we did not test for this in terms of mapped differences between different model outputs. Significance tests were conducted on time series comparisons between observations and model fields but were not commented on in explicitly in the original text. This was to keep the discussion clear with the emphasis here more on understanding the relative characteristics of the different simulation approaches (and noting there are many experiments considered), more so than quantitative/statistical comparison to observations. For example, for Titli,

the difference in bias relative to observations is statistically significant at 95% for all SST timeseries shown in Fig. 3(i)-(k) compared with the bias for ATMfix. This is also the case for all Fani SST timeseries (Fig. 4(i)-(k)), with exception of KPP at 23093 (Fig. 4(j)) for which the comparison to observations is only significantly different from ATMfix at 90% level. Statistical significance of results for wind speed relative to observations (Fig. 9) will be highlighted in the revised manuscript with AO and AOW having statistically significant improvement in wind speed bias relative to ATMfix control simulation.

Line 465-466. Could you please describe better how you detect and track tropical cyclone.
A reference to the tracking algorithm described by Heming (2017) was provided in line 464 and will be expanded in the updated manuscript, including details of the tuneable parameters used in the analysis presented in this paper. Section 3.3 of Heming (2017) gives further details of the method,.based on search within 3° radius of an observed cyclone centre of computed model maximum relative vorticity at 850 hPa and then minimum MSLP. Note this reference also gives some description of the basis for the observed track data used in this paper, based on synthesis from exchanged data from local centres.

I find really interesting the discussion and conclusions paragraph. Probably I missed the point but I do not understand if there exists a better configuration with respect to other tested in your experiments or which is able to balance different factors such as biases, computational time... Could you please infer a little bit more about?
Assessing if there are better configurations is part of ongoing research and development using RCS-IND1, so the configurations documented in this paper should be considered a baseline rather than 'final' optimised definition, with indeed the results presented showing some areas where the system can be improved, such as consistent use of frictional heating.
The discussion will be expanded slightly in the revised manuscript to be more specific on what the authors have in mind for future upgrades of model codes (i.e., to UMvn12+; NEMOvn4.2+; WWIIIvn7.2+) to be able to benefit from a breadth of community enhancements, as well as updating the regional atmosphere and land configuration used from RAL1-T to subsequent RAL2 (and RAL3 currently in development).
We also expand the bulled discussing computational performance, as indicated in Table 9, the current presentation reflects a system only 'optimised' to run in a suitable time for research purposes with no specific efforts taken to optimise load balancing between components, data volumes etc, to ensure the most efficient use of computational resources across different RUN_MODE options.
We trust these changes give a bit more of the detail that the reviewer is requested, while aiming to keep Section 4 at a reasonable length. It it useful to emphasise that development of the RCS-IND* configurations is ongoing, and the authors' intention to document further updates in subsequent documentation papers.

| Configuration | a | o | w | ak | aw | ao | aow |
|---|---|---|---|---|---|---|---|
| Nodes used | 48 | 15 | 10 | 49 | 58 | 63 | 73 |
| Runtime/day[1] | 17 min | 20 min | 5 min | 18 min | 18 min | 20 min | 22 min |
| Runtime/day[2] | 16 min | 21 min | | 16 min | | 24 min | |
| Output/day[3] (Diagnostic) | 20 Gb | 25 Gb | 2 Gb | 25 Gb | 22 Gb | 45 Gb | 47 Gb |
| Output/day[4] (Coupling) | 0 Gb | 0 Gb | 0 Gb | 26 Gb | 10 Gb | 51 Gb | 71 Gb |

*Revised* **Table 9. Summary of the typical computational resources required to run RCS-IND1 experiments, runtimes and output data volumes for completing a day simulation. Run durations quoted in row [1] were completed using the Met Office Cray XC40 and those completed in row labelled [2] were completed using the NCMRWF High Performance Computing Server Mihir Cray XC40. Two output data volume rows are given. The Diagnostic output (Row [3]) shows output data size saved to disk for daily restart and model variables of interest to enable analysis. Note number and type of outout diagnostics are dependent on user specifications, but values are indicative of default RCS-IND1 configurations and data volumes typically archived. The Coupling output (Row [4]) shows the volume of data written to disk to support coupling exchanges (computed as difference between total output volume quoted in daily log file and the Diagnostic output size on disk). Note the data volumes required for coupling are less user specific, and these data are not relevant for archiving, but will scale with choice of coupling frequency. All values reflect configurations without optimisation.**

[Figure]

***Revised* Figure 1: Illustration of RCS-IND1 domain coupled system domain extent. Shaded contours represent the atmosphere model orography over land and ocean model bathymetry respectively. Orography contours for land higher than 1000m (black shading) are marked with contours every 1000 m. The region highlighted by the maroon box shows the region of focus for results presented in this paper. Marked locations indicate in-situ observation points referred to in the results section: from north to south, Red = Gopalpur [84.9E, 19.3N]; Magenta = 23091 [89.2E, 17.8N]; Green = 23093 [88.0E, 16.3N]; Blue = 23459 [87.0E, 14.0N].**

---

## Referee Report (RR1)

**Title:** "The Regional Coupled Suite (RCS-IND1): application of a flexible regional coupled modelling framework to the Indian region at kmscale"

I appreciate the efforts the authors have done to address my concerns. Now I am happy to recommend this manuscript for publication in "Geoscientific Model Development".

**Recommendation:** Accept.

Thank you for answering my question on storm surge (RC1). I recommend adding your research plan on this topic in the **Key research priorities part (Line 750).** May be a little more detail on point 1.